# DiP Benchmark Tests: Evaluation Benchmarks for Discourse Phenomena in MT

## ABSTRACT

Despite increasing instances of machine translation (MT) systems including extrasentential context information, the evidence for translation quality improvement is sparse, especially for discourse phenomena. Popular metrics like BLEU are not expressive or sensitive enough to capture quality improvements or drops that are minor in size but significant in perception. We introduce the first of their kind MT benchmark testsets that aim to track and hail improvements across four main discourse phenomena: anaphora, lexical consistency, coherence and readability, and discourse connective translation. We also introduce evaluation methods for these tasks, and evaluate several competitive baseline MT systems on the curated datasets. Surprisingly, we find that the complex context-aware models that we test do not improve discourse-related translations consistently across languages and phenomena. Our evaluation benchmark is available as a leaderboard at <dipbenchmark1.github.io>.

## 1 INTRODUCTION AND RELATED WORK

The advances in neural machine translation (NMT) systems have led to great achievements in terms of state-of-the-art performance in automatic translation tasks. There have even been claims that their translations are no worse than what an average bilingual human may produce (Wu et al., 2016) or that the translations are on par with professional translators (Hassan et al., 2018). However, extensive studies conducting evaluations with professional translators (Läubli et al., 2018; Popel et al., 2020) have shown that there is a statistically strong preference for human translations in terms of fluency and overall quality when evaluations are conducted monolingually or at the document level.

Document (or discourse) level phenomena (*e.g.,* coreference, coherence) may not seem lexically significant, but contribute significantly to readability and understandability of the translated texts (Guillou, 2012). Targeted datasets for evaluating phenomena like coreference (Guillou et al., 2014; Guillou & Hardmeier, 2016; Lapshinova-Koltunski et al., 2018; Bawden et al., 2018; Voita et al., 2018b), or ellipsis and lexical cohesion (Voita et al., 2019), have been proposed.

The NMT framework such as the Transformer (Vaswani et al., 2017) provides more flexibility to incorporate larger context. This has spurred a great deal of interest in developing context-aware NMT systems that take advantage of source or target contexts, *e.g.,* Miculicich et al. (2018), Maruf & Haffari (2018), Voita et al. (2018b; 2019), Xiong et al. (2019), Wong et al. (2020), to name a few.

Most studies only report performance on specific testsets, often limited to improvements in BLEU (Papineni et al., 2002). Despite being the standard MT evaluation metric, BLEU has been criticised for its inadequacy; the scores are not interpretable, and are not sensitive to small improvements in lexical terms that may lead to big improvements in fluency or readability (Reiter, 2018). There is no framework for a principled comparison of MT quality beyond mere lexical matching as done in BLEU: there are no standard corpora and no agreed-upon evaluation measures.

To address these shortcomings, we propose the DiP benchmark tests (for **Di**scourse **P**henomena), that will enable the comparison of machine translation models across discourse task strengths and source languages. We create diagnostic testsets for four diverse discourse phenomena, and also propose automatic evaluation methods for these tasks. However, discourse phenomena in translations can be tricky to identify, let alone evaluate. A fair number of datasets proposed thus far have been manually curated, and automatic evaluation methods have often failed to agree with human

judgments (Guillou & Hardmeier, 2018). To mitigate these issues, we use trained neural models for identifying and evaluating complex discourse phenomena and conduct extensive user studies to ensure agreements with human judgments. Our methods for automatically extracting testsets can be applied to multiple languages, and find cases that are difficult to translate without having to resort to synthetic data. Moreover, our testsets are extracted in a way that makes them representative of current challenges. They can be easily updated to reflect future challenges, preventing the pitfall of becoming outdated, which is a common failing of many benchmarking testsets.

We also benchmark established MT models on these testsets to convey the extent of the challenges they pose. Although discourse phenomena can and do occur at the sentence-level (*e.g.,* between clauses), we would expect MT systems that model extra-sentential context (Voita et al., 2018b; Zhang et al., 2018; Miculicich et al., 2018) to be more successful on these tasks. However, we observe significant differences in system behavior and quality across languages and phenomena, emphasizing the need for more extensive evaluation as a standard procedure. We propose to maintain a leaderboard that tracks and highlights advances in MT quality that go beyond BLEU improvement.

Our main contributions in this paper are as follows:

- Benchmark testsets for four discourse phenomena: anaphora, coherence & readability, lexical consistency, and discourse connectives.
- Automatic evaluation methods and agreements with human judgments.
- Benchmark evaluation and analysis of four context-aware systems contrasted with baselines, for German/Russian/Chinese-English language pairs.

## 2    MACHINE TRANSLATION MODELS

**Model Architectures.**    We first introduce the MT systems that we will be benchmarking on our testsets. We evaluate a selection of established models of various complexities (simple sentence-level to complex context-aware models), taking care to include both source- and target-side context-aware models. We briefly describe the model architectures here:

- S2S: A standard 6-layer base Transformer model (Vaswani et al., 2017) which translates sentences independently.
- CONCAT: A 6-layer base Transformer whose input is two sentences (previous and current sentence) merged, with a special character as a separator (Tiedemann & Scherrer, 2017).
- ANAPH: Voita et al. (2018b) incorporate source context by encoding it with a separate encoder, then fusing it in the last layer of a standard Transformer encoder using a gate. They claim that their model explicitly captures *anaphora* resolution.
- TGTCON: To model target-context, we implement a version of ANAPH with an extra operation of multi-head attention in the decoder, computed between representations of the target sentence and target context. The architecture is described in detail in the Appendix (A.5).
- SAN: Zhang et al. (2018) use *source attention network*: a separate Transformer encoder to encode source context, which is incorporated into the source encoder and target decoder using gates.
- HAN: Miculicich et al. (2018) introduce a *hierarchical attention network* (HAN) into the Transformer framework to dynamically attend to the context at two levels: word and sentence. They achieve the highest BLEU when hierarchical attention is applied separately to both the encoder and decoder.

**Datasets and Training.**    The statistics for the datasets used to train the models are shown in Table 1. We tokenize the data using Jieba[1] for Zh and Moses scripts[2] for the other languages, lowercase the text, and apply BPE encodings[3] from Sennrich et al. (2016). We learn the BPE encodings with the command `learn-joint-bpe-and-vocab -s 40000`. The scores reported are BLEU4, computed either through **fairseq** or NLTK (Wagner, 2010). Further details about dataset composition, training settings and hyperparameters can be found in the Appendix (A.7).

---

[1] https://github.com/fxsjy/jieba
[2] https://www.statmt.org/moses/
[3] https://github.com/rsennrich/subword-nmt/

**BLEU scores.** The BLEU scores on the WMT-14 (De-En, Ru-En) and on the WMT-17 (Zh-En) testsets for each of the six trained models are shown in Table 2. We were unable to train HAN for Zh-En as the model was not optimized for training with large datasets. In contrast to increases in BLEU for selected language-pairs and datasets reported in published work, incorporating context within elaborate context-dependent models decreases BLEU scores for the Zh-En and De-En tasks. However, the simple concatenation-based model CONCAT performs better than S2S for De-En and Ru-En; this shows that context knowledge is indeed helpful for improving BLEU.

Table 1: Statistics for different language pairs showing the number of (parallel) sentences in the train/dev/test datasets. The test data is from WMT-14 for De-En and Ru-En, and WMT17 for Zh-En.

| Pair | Source | Train | Dev | Test |
|------|--------|-------|-----|------|
| **Zh-En** | IWSLT, Europarl, News, UN, WikiTitles | 17,195,748 | 2002 | 2001 |
| **De-En** | IWSLT, Europarl, News | 2,490,871 | 3,693 | 3,003 |
| **Ru-En** | IWSLT, News | 459,572 | 4,777 | 3,003 |

Table 2: BLEU scores achieved by benchmarked models on the WMT-14 (De-En, Ru-En) and the WMT-17 (Zh-En) testsets.

| Model | De-En | Ru-En | Zh-En |
|-------|-------|-------|-------|
| S2S | 31.65 | 23.88 | 17.86 |
| CONCAT | 31.96 | 24.56 | 17.17 |
| ANAPH | 29.94 | 27.66 | 16.31 |
| TGTCON | 29.94 | 26.06 | 15.67 |
| SAN | 29.32 | 24.34 | 15.18 |
| HAN | 29.69 | 25.11 | – |

Table 3: MT output errors indicative of source texts with hard-to-translate phenomena (WMT-19).

| Phenomena | Source | MT output | Reference |
|-----------|--------|-----------|-----------|
| Anaphora | Der französische Busfahrer war zur Krankenhaus geblieben. Er werde im Laufe des Tages entlassen. | The French bus driver had remained in the hospital. It will be released over the day. | The French bus driver had stayed in the hospital. He will be released over the day. |
| Coherence & Readability | В Ростове внедорожник перевернулс-я после столкновения с автомобилем такси. Происшествие случилось 29 сентября на улице Немировича -Данченко. | In Rostov, the SUV turned over after collision with a taxi car. The procession occurred on September 29 in Nemirovic Street, Dunchenko. | SUV flips over after collision with taxi in Rostov. The accident took place on September 29 on Nemirovich-Danchenko street. |
| Lexical Consistency | *Bauen in Landsham* *Grundstücke in Landsham-Süd* *Neubaugebiet Landsham-Süd* | *building in landsham* *land in landsh-south* *area of landsh-süd* | *building in landsham* *properties in south landsham* *south landsham area* |
| Discourse Connectives | 让世界看到开放市场、共享未来的中国自信和中国担当。独行快，众行远。 | ..share the future of China's self-confidence and China's commitment. Walk Alone fast, the crowd goes far. | ..shows the world China's confidence and responsibility to open its markets and share its future. It is fast to go alone but it is further to go in crowds. |

# 3 BENCHMARK TESTSETS

We construct our benchmarking testsets based on four main principles:

**Selectivity.** The testsets need to provide hard to translate contexts for MT models. We ensure this by looking at translation errors made by system submissions to campaigns like WMT and IWSLT.

**Authenticity.** The testsets cannot contain artificial or synthetic data but only natural text. Rather than generating testset samples using heuristics, we extract hard contexts from existing human-generated source text.

**Multilinguality.** The testset extraction method should be automatic and applicable to multiple languages. Our framework can be used to extract testsets for all source languages that are part of the considered MT campaigns.

**Adaptability.** The testsets should be easy to update frequently, making them adaptable to improvements in newer systems. Since we automatically extract hard contexts based on MT errors, our testsets are easy to update; they adapt to errors in newer (and possibly more accurate) systems, making the tasks harder over time.

We use the system outputs released by WMT and IWSLT for the most recent years (Nadejde et al., 2016; Bojar et al., 2017; 2018; 2019; Cettolo et al., 2016; 2017) to build our testsets. For De-En,

Ru-En and Zh-En, these consist of translation outputs from 68, 41 and 47 unique systems respectively. Since the data comes from a wide variety of systems, our testsets representatively aggregate different types of errors from several (arguably SOTA) models. Also note that the MT models we are benchmarking are not a part of these system submissions to WMT, so there is no potential bias in the testsets.

## 3.1 ANAPHORA

Anaphora are references to entities that occur elsewhere in a text; mishandling them can result in ungrammatical sentences or the reader inferring the wrong antecedent, leading to misunderstanding of the text (Guillou, 2012). We focus specifically on the aspect of incorrect *pronoun* translations.

**Testset.** To obtain hard contexts for pronoun translation, we look for source texts that lead to erroneous pronoun translations in system outputs. We align the system translations with their references, and collect the cases in which the translated pronouns do not match the reference.[4]

Our anaphora testset is an updated version of the one proposed by Jwalapuram et al. (2019). We filter the system translations based on their list of cases where the translations can be considered wrong, rather than acceptable variants. The corresponding source texts are extracted as a test suite for pronoun translation. This gives us a pronoun benchmark testset of 2564 samples for De-En, 2368 for Ru-En and 1540 for Zh-En.

**Evaluation.** Targeted evaluation of pronouns in MT has been challenging as it is not fair to expect an exact match with the reference. Evaluation methods like APT (Miculicich Werlen & Popescu-Belis, 2017) or AutoPRF (Hardmeier & Federico, 2010) are specific to language pairs or lists of pronouns, requiring extensive manual intervention. They have also been criticised for failing to produce evaluations that are consistent with human judgments (Guillou & Hardmeier, 2018).

Jwalapuram et al. (2019) propose a pairwise ranking model that scores "good" pronoun translations (like in the reference) higher than "poor" pronoun translations (like in the MT output) in context, and show that their model is good at making this distinction, along with having high agreements with human judgements. However, they do not rank multiple system translations against each other, which is our main goal; the absolute scores produced by their model are not useful since it is trained in a pairwise fashion.

We devise a way to use their model to score and rank system translations in terms of pronouns. First, we re-train their model with more up-to-date WMT data (more details in Appendix A.1). We obtain a score for each benchmarked MT system (S2S, CONCAT, etc.) translation using the model, plus the corresponding reference sentence. We then *normalize* the score for each translated sentence by calculating the difference with the reference. To get an overall score for an MT system, the assigned scores are summed across all sentences in the testset.

$$\text{Score}_{\text{sys}} = \sum_i \rho_i(\text{ref}|\theta) - \rho_i(\text{sys}|\theta) \tag{1}$$

where $\rho_i(.|\theta)$ denotes the score given to sentence $i$ by the pronoun model $\theta$. The systems are ranked based on this overall score, where a lower score indicates a better performance. We conduct a user study to confirm that the model rankings correspond with human judgments, obtaining an agreement of **0.91** between four participants who annotated 100 samples. Appendix A.1 gives details (*e.g.,* interface, participants, agreement) about the study.

### 3.1.1 RESULTS AND ANALYSIS

The ranking results obtained from evaluating the MT systems on our pronoun benchmark testset using our evaluation measure are given in Table 4 (first two columns). We also report common pronoun errors for each model based on our manual analysis (last three columns). Specifically, we observed the following types of errors in our analysis of a subset of the translation data:

(*i*) **Gender copy.** Translating from De/Ru to En often requires 'flattening' of gendered pronouns to *it*, since De/Ru assign gender to all nouns. In many cases, machine translated pronouns tend to (mistakenly) agree with the source language. For example, ***diese Wohnung*** *in Earls Court..., und **sie***

---

[4]This process requires the pronouns in the target language to be separate morphemes, as in English.

Table 4: Pronoun evaluation: **Rank**ings of the different models for each language pair, obtained from our evaluation procedure. Through manual analysis, % for the following types of errors are reported: Anaphora - instances of **G**ender **C**opy, **N**amed **E**ntity and **Lang**uage specific errors.

| Rank | Model | GC | NE | Lang | Rank | Model | GC | NE | Lang | Rank | Model | GC | NE | Lang |
|------|-------|----|----|------|------|-------|----|----|------|------|-------|----|----|------|
| | **De-En** | | | | | **Ru-En** | | | | | **Zh-En** | | | |
| 1 | CONCAT | 55 | 33 | 11 | 1 | HAN | 31 | 48 | 21 | 1 | ANAPH | 0 | 80 | 20 |
| 2 | S2S | 63 | 25 | 12 | 2 | ANAPH | 29 | 46 | 25 | 2 | CONCAT | 0 | 25 | 75 |
| 3 | SAN | 27 | 27 | 46 | 3 | CONCAT | 29 | 46 | 25 | 3 | S2S | 0 | 40 | 60 |
| 4 | HAN | 44 | 22 | 33 | 4 | SAN | 32 | 44 | 24 | 4 | TGTCON | 0 | 10 | 90 |
| 5 | ANAPH | 42 | 17 | 41 | 5 | S2S | 37 | 37 | 26 | 5 | SAN | 0 | 33 | 67 |
| 6 | TGTCON | 0 | 20 | 80 | 6 | TGTCON | 17 | 8 | 75 | | | | | |

*hatte...* is translated to : ***apartment** in Earls Court, and **she** had...*, which keeps the female gender expressed in *sie*, instead of translating it to *it*.

(*ii*) **Named entity.** A particularly hard problem is to infer gender from a named entity, *e.g.,* **Lady Liberty**...***She** is meant to...*- **she** is wrongly translated to **it**. Such examples demand higher inference abilities (*e.g.,* distinguish male/female names).

(*iii*) **Language specific phenomena.** In Russian and Chinese, pronouns are often dropped - sentences become ungrammatical in English without them. Pronouns can also be ambiguous in the source language; *e.g.,* in German, the pronoun *sie* can mean both *she* and *you*, depending on capitalization, sentence structure, and context.

Overall, we observe that the advantages of contextual models are not consistent across languages. They seem to use context well in Ru-En, but fail to outperform S2S or CONCAT in De-En, while Zh-En is inconclusive. The TGTCON model is consistently poor in this task. The partial success of the S2S model can be explained by its tendency to use *it* as the default pronoun, which statistically appears most often due to the lack of grammatical gender in English. More variability in pronouns occurs in the outputs of the context-aware models, but this does not contribute to a greater success.

## 3.2 COHERENCE AND READABILITY

Pitler & Nenkova (2008) define coherence as the ease with which a text can be understood, and view readability as an equivalent property that indicates whether it is well-written.

**Testset.** To test for coherence and readability, we try to find documents that can be considered hard to translate. We use the coherence model proposed by Moon et al. (2019), which is trained in a pairwise ranking fashion on WSJ articles, where a negative document is formed by shuffling the sentences of an original (positive) document. It models syntax, inter-sentence coherence relations and global topic structures. It has been shown in some studies that MT outputs are incoherent (Smith et al., 2015; 2016; Läubli et al., 2018). We thus re-train the coherence model with reference translations as positive and MT outputs as negative documents to better capture the coherence issues that are present in MT outputs (more details in Appendix A.2). We use older WMT submissions from 2011-2015 to ensure that the training data does not overlap with the benchmark testset data.

The coherence model takes a system translation (multi-sentential) and its reference as input and produces a score for each. Similar to Eq. 1, we consider the difference between the scores produced by the model for the reference and the translated text as the coherence score for the translated text.

For a given source text (document) in the WMT testsets, we obtain the coherence scores for each of the translations (*i.e.,* WMT/IWSLT submissions) and average them. The source texts are sorted based on the average coherence scores of their translations. The texts that have lower average coherence scores can be considered to have been hard to translate coherently. We extract the *source* texts with scores below the median. These source texts form our benchmark testset for coherence and readability. This yields 272 documents (5,611 sentences) for De-En, 330 documents (4,427 sentences) for Ru-En and 210 documents (3,050 sentences) for Zh-En.

**Evaluation.** Coherence and readability is also a hard task to evaluate, as it can be quite subjective. We resort to model-based evaluation here as well, to capture the different aspects of coherence in translations. We use our re-trained coherence model to score the benchmarked MT system translations and modify the scores for use in the same way as the anaphora evaluation (Eq. 1) to obtain a

relative ranking. As mentioned before (Sec. 3), the benchmarked MT systems do not overlap with the WMT system submissions, so there is no potential bias in evaluation since the testset extraction and the evaluation processes are independent. To confirm that the model does in fact produce rankings that humans would agree with, and to validate our model re-training, we conduct a user study, and obtain an agreement of **0.82** between three participants who annotated 100 samples. More details about the study can be found in the Appendix (A.2).

### 3.2.1 RESULTS AND ANALYSIS

We identified some frequent coherence and readability errors (more examples in Appendix A.8):

***(i)* Inconsistency.** As in (Somasundaran et al., 2014), we observe that inconsistent translation of words across sentences (in particular named entities) breaks the continuity of meaning.

***(ii)* Translation error.** Errors at various levels spanning from ungrammatical fragments to model hallucinations introduce phrases which bear little relation to the whole text (Smith et al., 2016):

Table 5: Coherence and Readability evaluation: **R**ankings of the different models for each language pair, obtained from our evaluation procedure.

| Rk | De-En | Ru-En | Zh-En |
|----|-------|-------|-------|
| 1 | CONCAT | ANAPH | ANAPH |
| 2 | SAN | CONCAT | CONCAT |
| 3 | S2S | TGTCON | S2S |
| 4 | ANAPH | SAN | TGTCON |
| 5 | TGTCON | S2S | SAN |
| 6 | HAN | HAN | - |

Reference: *There is huge applause for the Festival Orchestra, who appear on stage for the first time – in casual leisurewear in view of the high heat.*
Translation: *There is great applause for the solicitude orchestra , which is on the stage for the first time, with the heat once again in the wake of an empty leisure clothing.*

From the rankings in Table 5, we can see that ANAPH is the most coherent model for Zh-En and Ru-En but performs poorly in De-En, similar to the pronoun benchmark. Generally CONCAT is better than complex contextual models in this task.

### 3.3 LEXICAL CONSISTENCY

Lexical consistency in translation was first defined as 'one translation per discourse' by Carpuat (2009), *i.e.,* the translation of a particular source word consistently to the same target word in that context. Guillou (2013) analyze different human-generated texts and conclude that human translators tend to maintain lexical consistency to support the important elements in a text. The consistent usage of lexical items in a discourse can be formalized by computing the *lexical chains* (Morris & Hirst, 1991; Lotfipour-Saedi, 1997).

**Testset.** To extract a testset for lexical consistency evaluation, we first align the translations from the system submissions with their references. In order to get a reasonable lexical chain formed by a consistent translation, we consider translations of blocks of 3-5 sentences in which the (lemmatized) word we are considering occurs at least twice in the reference. For each such word, we check if the corresponding system translation produces the same (lemmatized) word at least once, but fewer than the number of times the word occurs in the reference. In such cases, the system translation has failed to be lexically consistent in translation (see Table 3 for an example). We limit the errors considered to nouns and adjectives. The source texts of these cases form the benchmark testset. This gives us a testset with 618 sets (*i.e.,* text blocks) for De-En (3058 sentences), 732 sets for Ru-En (3592 sentences) and 961 sets for Zh-En (4683 sentences).

**Evaluation.** For lexical consistency, we adopt a simple evaluation method. For each block of 3-5 sentences, we either have a consistent translation of the word in focus, or the translation is inconsistent. We simply count the instances of consistency and rank the systems based on the percentage. Model translations are considered lexically inconsistent if at least one translation of a particular word matches the reference translation, but this translated word occurs fewer times than in the reference. For samples where no translations match the reference, we cannot be sure about inconsistency, since a synonym of the reference translation could have been used consistently. Therefore, we do not consider them for calculating the percentage used for the main ranking, but we report the consistency percentage as a fraction of the full testset for comparison (further discussion in Appendix (A.3)).

Table 6: Lexical consistency evaluation: **R**ankings of the different models for each language pair, ranked by the % of samples that are **Con**sistent compared to inconsistent samples, then by % of consistent samples in the **Full** testset. Also shown are the results of manual error analysis on a subset of the translations for **Syn**onyms, **Rel**ated words, **Om**issions, **N**amed **E**ntity, **Rand**om translation.

| | De-En | | | | | | | | | Ru-En | | | | | | |
|---|---|---|---|---|---|---|---|---|---|---|---|---|---|---|---|---|
| Rk | Model | %Con | %Full | Syn | Rel | Om | NE | Rd | Rk | Model | %Con | %Full | Syn | Rel | Om | NE | Rd |
| 1 | CONCAT | 42.83 | 34.30 | 38 | 15 | 23 | 4 | 19 | 1 | TGTCON | 26.36 | 11.88 | 20 | 8 | 28 | 28 | 16 |
| 2 | TGTCON | 42.37 | 32.36 | 13 | 27 | 7 | 27 | 27 | 2 | ANAPH | 23.99 | 12.16 | 15 | 0 | 26 | 15 | 44 |
| 3 | SAN | 41.52 | 30.90 | 38 | 19 | 24 | 5 | 14 | 3 | CONCAT | 23.81 | 12.29 | 15 | 8 | 15 | 18 | 44 |
| 4 | ANAPH | 40.72 | 30.90 | 46 | 21 | 21 | 4 | 8 | 4 | SAN | 19.22 | 9.42 | 6 | 9 | 24 | 18 | 42 |
| 5 | S2S | 40.00 | 32.03 | 38 | 19 | 29 | 5 | 9 | 5 | S2S | 18.99 | 9.29 | 21 | 9 | 27 | 21 | 21 |
| 6 | HAN | 38.41 | 29.77 | 35 | 22 | 30 | 4 | 7 | 6 | HAN | 17.44 | 8.19 | 11 | 8 | 19 | 19 | 41 |

| | Zh-En | | | | | | | |
|---|---|---|---|---|---|---|---|---|
| Rk | Model | %Con | %Full | Syn | Rel | Om | NE | Rd |
| 1 | ANAPH | 30.94 | 17.48 | 43 | 14 | 29 | 0 | 14 |
| 2 | TGTCON | 29.84 | 16.02 | 14 | 0 | 29 | 43 | 14 |
| 3 | S2S | 28.27 | 18.21 | 0 | 25 | 38 | 25 | 13 |
| 4 | CONCAT | 28.17 | 16.65 | 0 | 66 | 0 | 16 | 33 |
| 5 | SAN | 26.33 | 13.42 | 0 | 13 | 38 | 25 | 25 |

### 3.3.1 RESULTS AND ANALYSIS

The rankings of the MT systems based on the percentage of samples with consistent translations on the lexical consistency benchmark testsets are given in Table 6 (first four columns), along with our findings from a manual analysis on a subset of the translations (last five columns). Our manual inspection of the lexical chains shows the following tendencies:

(*i*) **Synonyms & related words.** Words are exchanged for their synonyms (*poll - survey*), hypernyms/hyponyms (*ambulance - car*) or related concepts (*wine - vineyard*).

(*ii*) **Named entities.** Models tend to distort proper names and translate them inconsistently. For example, the original name *Füchtorf* (name of a town) gets translated to *feeding-community*.

(*iii*) **Omissions.** Occurs when words are omitted altogether from the lexical chain.

The overall low quality of Russian translations contributes to the prevalence of **Rand**om translations, and the necessity to transliterate named entities increases **NE** errors for both Ru-En and Zh-En. Here we see some complex contextual models performing well; TGTCON leads the board across De-En, Ru-En and Zh-En, with ANAPH performing similarly well for Ru-En and Zh-En. Generally, we should be seeing a consistent advantage for target-side context models, which should be able to "remember" their own translation of a word from previous sentences; however this only materializes for TGTCON and not for HAN.

### 3.4 DISCOURSE CONNECTIVES

Discourse connectives are used to link the contents of texts together by signaling coherence relations that are essential to the understanding of the texts (Prasad et al., 2014). Failing to translate a discourse connective correctly can result in texts that are hard to understand or ungrammatical. Finding errors in discourse connective translations can be quite tricky, since there are often many acceptable variants. To mitigate confusion, we limit the errors we consider in discourse connectives to the setting where the reference contains a connective but the translations fail to produce any (see Table 3 for an example).

**User Study.** To confirm that missing connectives are problematic, we conduct a user study. Participants are shown two previous sentences from the reference for context, and asked to choose between two candidate options for the sentence that may follow. These options consist of the reference translation which includes a connective, and an MT output that is missing the connective translation. Participants are asked to choose the sentence which more accurately conveys the intended meaning. See Figure 4b in Appendix A.4 for an example interface.

We obtain an agreement of **0.82** between two participants who annotated 200 samples, that translations with connectives are preferred. If the MT outputs with missing connectives were structured in such a way as to have implicit discourse relations, the agreements that favoured the references should be significantly lower. However, the strong agreements favouring the reference with the con-

Table 7: Discourse connective evaluation: **Rank**ings of the different models for each language pair, ranked first by their **Acc**uracy and then by the percentage where **ANY** connective is produced. Each set of rankings is followed by the results of the manual analysis on a subset of the translation data for **Om**issions, **Syn**onyms, **Mis**translations.

| | De-En | | | | | | | | Ru-En | | | | | | | | Zh-En | | | | | |
|---|---|---|---|---|---|---|---|---|---|---|---|---|---|---|---|---|---|---|---|---|---|---|
| Rank | Model | Acc | ANY | Om | Syn | Mis | | Rank | Model | Acc | ANY | Om | Syn | Mis | | Rank | Model | Acc | ANY | Om | Syn | Mis |
| 1 | Anaph | 49.42 | 75.58 | 75 | 25 | 0 | | 1 | Anaph | 40.81 | 68.70 | 63 | 30 | 7 | | 1 | S2S | 55.24 | 80.66 | 43 | 42 | 14 |
| 2 | San | 48.25 | 72.67 | 67 | 33 | 0 | | 2 | S2S | 37.41 | 73.47 | 59 | 28 | 12 | | 2 | TgtCon | 52.48 | 82.04 | 44 | 26 | 30 |
| 3 | TgtCon | 48.25 | 72.09 | 40 | 53 | 6 | | 3 | TgtCon | 36.05 | 63.95 | 73 | 19 | 8 | | 3 | San | 49.17 | 79.28 | 44 | 33 | 22 |
| 4 | S2S | 47.67 | 73.84 | 76 | 24 | 0 | | 4 | San | 35.37 | 60.54 | 62 | 28 | 9 | | 4 | Anaph | 48.62 | 77.90 | 57 | 29 | 14 |
| 5 | Concat | 44.77 | 70.93 | 68 | 32 | 0 | | 5 | Concat | 32.65 | 70.06 | 61 | 32 | 6 | | 5 | Concat | 48.06 | 76.79 | 20 | 40 | 40 |
| 6 | Han | 44.18 | 69.76 | 72 | 28 | 0 | | 6 | Han | 31.29 | 55.10 | 76 | 21 | 3 | | | | | | | | |

nective indicate that the missing connectives in MT outputs are indeed an issue. More details about the study can be found in the Appendix (A.4).

**Testset.** It would not be appropriate to simply extract connectives using a list of candidates, since those words may not always act in the capacity of a discourse connective. In order to identify the discourse connectives, we build a simple explicit connective classifier (a neural model) using annotated data from the Penn Discourse Treebank or PDTB (Prasad et al., 2018) (details in Appendix A.4). The classifier achieves an average cross-validation F1 score of **93.92** across the 25 sections of PDTBv3, proving that it generalizes well.

After identifying the explicit connectives in the reference translations, we align them with the corresponding system translations and extract the source texts of cases with missing connective translations. We only use the classifier on the reference text, but consider all possible candidates in the system translations to give them the benefit of the doubt. This gives us a discourse connective benchmark testset with 172 samples for De-En, 147 for Ru-En and 362 for Zh-En.

**Evaluation.** There has been some work on semi-automatic evaluation of translated discourse connectives (Meyer et al., 2012; Hajlaoui & Popescu-Belis, 2013); however, it is limited to only En-Fr, based on a dictionary list of equivalent connectives, and requires using potentially noisy alignments and other heuristics. In the interest of evaluation simplicity, we expect the model to produce the same connective as the reference. Since the nature of the challenge is that connectives tend to be omitted altogether, we report both the accuracy of connective translations with respect to the reference, and the percentage of cases where *any* candidate connective is produced.

### 3.4.1 RESULTS AND ANALYSIS

The rankings of MT systems based on their accuracy of connective translations are given in Table 7, along with our findings from a manual analysis on a subset of the translations. In benchmark outputs, we observed mostly **omissions** of connectives (disappears in the translation), **synonymous translations** (*e.g., Naldo is also a great athlete on the bench - Naldo's "great sport" on the bank, too.*), and **mistranslations**. More examples can be found in the Appendix (A.8).

The ranking shows that the S2S model performs well for Ru-En and Zh-En but not for De-En. ANAPH continues its high performance in Ru-En and this time also De-En, doing poorly for Zh-En, while HAN is consistently poor with a lot of omissions.

## 4 DISCUSSION

Our benchmarking re-emphasizes the gap between BLEU scores and translation quality at the discourse level. The overall BLEU scores for De-En and Ru-En are higher than the BLEU scores for Zh-En; however, we see that Zh-En models have higher accuracies in the discourse connective task, and also outperform Ru-En in lexical consistency. Similarly, for Ru-En, both SAN and HAN have higher BLEU scores than the S2S and CONCAT models, but are unable to outperform these simpler models consistently in the discourse tasks, often ranking last.

We also reveal a gap in performance consistency across language pairs. Models may be tuned for a particular language pair, such as ANAPH trained for En-Ru (Voita et al., 2018a). For the same language pair (Ru-En), we show results consistent with what is reported; the model ranks first or second for all phenomena. However, it is not consistently successful in other languages, e.g., ranking

close to bottom for almost all cases in De-En. In general, our findings match the conclusions from Kim et al. (2019) regarding the lack of satisfactory performance gains in context-aware models.

Although our testsets and evaluation procedures have their limitations, like only checking for missing connectives or being unable to detect consistently translated synonyms of reference translations, they are a first step toward a standardized, comprehensive evaluation framework for MT models that spans multiple languages. They are useful for measuring basic model proficiency, performance consistency and for discovering MT deficiencies. Discourse-aware models have been advocated for improving MT (Sennrich, 2018); as more models are proposed, our framework will be a valuable resource that provides a better picture of model capabilities. With advances in NMT models and also in evaluation models for complex phenomena, harder challenges can be added and evaluated.

## 5 GENERALIZABILITY TO OTHER LANGUAGES

Procedures used to create our testsets can be generalized to create testsets in other languages. We briefly describe the possibilities here:

- **Anaphora:** The pronouns need to be separate morphemes (and not attached to verbs etc.). If there are several equivalent pronoun translations, a list may be needed so they can be excluded from being considered translation errors; *e.g.,* Miculicich Werlen & Popescu-Belis (2017) has such a list for French, a list can also be collected through user studies as in Jwalapuram et al. (2019).
- **Coherence & Readability:** The coherence model (Moon et al., 2019) used to find poorly translated texts was re-trained on reference vs. MT outputs. It is also possible to do this for other languages for which WMT (or IWSLT) system outputs are available. The coherence model from Moon et al. (2019) is an end-to-end neural model that does not rely on any language-specific features, and thus can be trained on any target language. However, language-specific or multilingual coherence models could also be used since Moon et al. (2019) primarily train and test their model on English (WSJ) data.
- **Lexical Consistency:** A lemmatizer was used to reduce common suffixes for detecting lexical consistency (*e.g.,* "box" and "boxes" should not be detected as inconsistent words), so a similar tool will be needed for any other target language; *e.g.,* CLTK (Johnson et al., 2014–2020) provides a lemmatizer for several languages.
- **Discourse Connectives:** Discourse connectives also need to be separate morphemes. We built a classifier trained on PDTB data to identify connectives since they are ambiguous in English. Datasets analogous to PDTB in other languages *e.g.,* PCC (German) (Bourgonje & Stede, 2020) and CDTB (Chinese) (Zhou & Xue, 2015), etc. are available.

## 6 CONCLUSIONS

We presented the first of their kind discourse phenomena based benchmarking testsets called the DiP tests, designed to be challenging for NMT systems. Our main goal is to emphasize the need for comprehensive MT evaluations across phenomena and language pairs, which we do by highlighting the performance inconsistencies of complex context-aware models. We will release the discourse benchmark testsets and evaluation frameworks for public use, and also propose to accept translations from MT systems to maintain a leaderboard for the described phenomena.

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

# A  APPENDIX

## A.1 ANAPHORA

**Source: German**
Der Siegeszug von Laura Siegemund endete im Viertelfinale von Rio: Die Weltranglisten-32. aus Metzingen verpasste durch ein 1:6, 1:6 gegen Monica Puig (Puerto Rico/WTA-Nr. 34) das Halbfinale des olympischen Tennisturniers deutlich. **Siegemund, die von Rückenproblemen geplagt wurde, konnte ihren Aufschlag nur einmal durchbringen und wirkte im Vergleich zu den vorherigen Runden kraftlos.**

**Candidate A:**
Siegemund, **who** had been plagued by back problems, was able to get **its** surcharge only once and was powerless in comparison to the previous rounds.

`1 ▾`

**Candidate B:**
Siegemund, **who** was plagued by back problems, could only get through to **their** strike and had a powerful effect compared to the previous rounds.

`1 ▾`

**Candidate C:**
Sigemund, **who** has been struggling with back problems, was only able to hold **her** serve once, and seemed powerless in comparison to earlier rounds.

`1 ▾`

Figure 1: User study interface (bilingual setup) for pronoun translation ranking. Pronouns in the sentences are highlighted in red.

**Context:**
The next day, they set off on the section towards St. Peter, the most demanding section, seeing them climb around 500 metres in altitude in sweltering heat, and then descend again. They earned the reward of barbecueing together by the swimming pond.

**Candidate A:**
After the morning impulse on the fourth day of pilgrimage in the imposing monastery church of St. Peter, **it** went to the final stage to Freiburg.

`1 ▾`

**Candidate B:**
After the morning impulse on the fourth pilgrimage day in the impressive monastery church of St. Peter, **we** went to the final stage to Freiburg.

`1 ▾`

**Candidate C:**
After the morning session on the fourth day of the pilgrimage in the impressive cloisters of St. Peter, **they** started off on the final stage to Freiburg.

`1 ▾`

Figure 2: User study interface (monolingual setup) for pronoun translation ranking. Pronouns in the sentences are highlighted in red.

**Detailed Steps for Testset Generation.** Steps that were used to generate the anaphora test set:

1. We aligned the references with various MT system outputs from WMT & IWSLT campaigns.

2. Given a list of pronouns (a closed set), we find the cases where the pronoun translations in the references do not match the aligned, corresponding pronoun translations in the MT system outputs.

3. Jwalapuram et al. (2019) provide a list of pronoun pairs that are not equivalent (non-interchangeable) translations of each other. We filter out the cases where the mismatched pronoun translations could be equivalent translations.

4. The corresponding source texts of the remaining wrongly translated cases are added to our test set.

**Evaluation Model Description.** Jwalapuram et al. (2019) build a pairwise ranking model that is trained to distinguish "good" pronoun translations (like in the reference) from "bad" pronoun translations (like in erroneous MT outputs) in context. We briefly describe the model architecture:

1. The model first obtains contextual representations of a sentence (reference or MT output) by attending over a common context of two previous sentences from the reference.

2. The pronoun representations from this sentence are extracted, and these in turn attend over the contextual sentence representations. This results in contextual representations of the pronouns in the sentence.

3. These contextual pronoun representations are passed on to the final layer which generates the scores.

4. The model is trained with a pairwise ranking objective to score the reference pronoun representations higher than the erroneous MT pronoun representations.

We believe that the model scores are both sensitive and specific to pronouns, for several reasons:

- **Sensitivity.** Results from the original paper are reported for:
  - Reference vs. MT output, where the model is 90.69% accurate.
  - Reference vs. noisy reference, where everything in the noisy version is identical except for the pronoun; here the accuracy is 89.11%.
  - High agreements (>=0.8) with human judgments on similar noisy data.
- **Specificity.** The model scores only originate from pronouns because:
  - The final layer of the model only uses pronoun representations to generate the scores.

Table 8: Results of the re-trained pronoun scoring model.

| Training data | Test data | Accuracy |
|---|---|---|
| WMT13-18 | WMT-19 | 86.76 |

— Attention heat maps from Jwalapuram et al. (2019) show that the scores are influenced only by the pronouns in the sentence, both for the reference vs. noisy and the reference vs. system translation cases.
— We also conduct a user study to confirm agreement between the model scores and human judgments, which is described later in this section.

**Re-trained model.**    The pronoun evaluation model results reported in Jwalapuram et al. (2019) are based on a model that is trained on WMT11-15 data and tested on WMT-2017 data. We re-train the model with more up-to-date data from WMT13-18, and test the model on WMT-19 data. Note that this training data is taken from WMT submissions, which do not overlap with the benchmarked MT models; there is therefore no conflict in using this trained model to evaluate the benchmarked model translations. Results are shown in Table 8. Their model scores the translations in context; we provide the previous two sentences from the reference translation as context according to their settings.

**User study.**    To confirm that our normalization-based ranking of systems agrees with human judgments, we conducted a user study. Participants are asked to rank given translation candidates in terms of their pronoun usage. We include the reference in the candidates, as a control.

We ask participants to rank system translations directly rather than a synthetically constructed contrastive pair (as was done by Jwalapuram et al. (2019)) to ensure that our evaluations, which will be conducted on actual translated texts, are reliable.

We first conducted the study in a *bilingual* setup, in the presence of the source for German-English. Participants were shown a source context of two sentences and the source sentence in bold, followed by three candidate translations of the source sentence, one of which is the reference. The other two were translations with different pronoun errors produced by MT systems. 3 participants annotated 100 such samples. The bilingual (German-English) user study interface for pronoun translation ranking is shown in Figure 1.

We then conducted the study in a *monolingual* setup, *i.e.,* native speakers are shown the reference context in English, and the two candidate English translations and the reference translation as possible options for the sentence that follows (Figure 2); 4 participants annotated 100 such samples.

The results are analysed to check how often (*i*) the *reference is preferred over the system translations* (our control), and (*ii*) the users agree in *preference over the system translations* (*i.e.,* human judgment for translation quality).

Due to the nature of the dataset, annotators are more likely to choose the reference as the better candidate, which yields a skewed distribution of the annotations; traditional correlation measures such as Cohen's kappa are not robust to this, and thus for this and all subsequent studies, we report the more appropriate Gwet's AC1/gamma coefficient. It is also the agreement reported by Jwalapuram et al. (2019). The control yielded a Gwet's AC1 agreement of **0.72** for the bilingual study and **0.82** for the monolingual study. The agreements are higher for the monolingual study; Läubli et al. (2018) also find that human judgements differ in monolingual setups.

We evaluate the rankings from our modified evaluation method with the monolingual study data and obtain an agreement of **0.91**, justifying the use of our modified pronoun model for evaluation.

**Results.**    The total assigned scores (difference between the reference and translation scores) obtained for each system after summing the over the samples in the respective testsets are given in Table 9. The models are ranked based on these scores from lowest score (least difference compared to reference; best performing) to highest score (highest difference compared to reference; worst performing).

Table 9: Models **ranked** according to their performance (best to worst) in anaphora according to the evaluation model, with BLEU score for comparison. Model scores given here are obtained by subtracting the score for the model translation from the score for the reference translation, and summing the score differences across the dataset. Hence, smaller model scores indicate better performance (closer to reference scores).

| De-En | | | | Ru-En | | | | Zh-En | | | |
|---|---|---|---|---|---|---|---|---|---|---|---|
| Rk | Model | BLEU | Score | Rk | Model | BLEU | Score | Rk | Model | BLEU | Score |
| 1 | CONCAT | 31.96 | 112.583 | 1 | HAN | 25.11 | 160.411 | 1 | ANAPH | 16.31 | 43.930 |
| 2 | S2S | 31.65 | 113.783 | 2 | ANAPH | 27.66 | 164.603 | 2 | CONCAT | 17.17 | 49.092 |
| 3 | SAN | 29.32 | 117.838 | 3 | CONCAT | 24.56 | 168.092 | 3 | S2S | 17.86 | 64.176 |
| 4 | HAN | 29.69 | 118.067 | 4 | SAN | 24.34 | 176.143 | 4 | TGTCON | 15.76 | 111.683 |
| 5 | ANAPH | 29.94 | 129.662 | 5 | S2S | 23.88 | 181.887 | 5 | SAN | 15.18 | 123.908 |
| 6 | TGTCON | 29.94 | 131.699 | 6 | TGTCON | 26.06 | 183.969 | | | | |

**Candidate A:**
Great Britain's track cycling team breaks the Olympic record twice and wins in sprint. The British track cycling team broke the Olympic record twice on Thursday and won the match sprint for men. The British cyclists broke the first of the records in qualification and then it was broken again in the first round of the race by New Zealand athletes. The Great Britain and New Zealand teams were racing in the final and the British athletes took away the title of Olympic record holder from New Zealanders, leaving them with silver medals.

[1 ▾]

**Candidate B:**
The British track cycling team has twice the Olympic record and wins the sprint. The British track team corrected the Olympic record twice on Thursday and won a sprint race for men. The first Olympic record was corrected by the British, but in the first round the competition was recorrected by New Zealanders. The British team with New Zealand met in the sprint finale, and the British took the title of the Olympic record owners to the young, leaving them with silver medals.

[1 ▾]

**Candidate C:**
The British track cycling team Olympic record twice right and wins the race. The British track team Thursday twice the Olympic record and edit won the sprint race at the men. The first Olympic record British qualifications, but repairing a race in the first round it was fixing the New Zealanders. Sprint finals in the British team to New Zealand, and the British said the New Zealanders in the Olympic record holder of the title, leaving them with silver medals.

[1 ▾]

Figure 3: User interface for coherence study. The participants are shown 4-sentence texts and asked to rank them in terms of coherence and readability.

## A.2 COHERENCE

**Re-trained model.** We re-train the pairwise coherence model in Moon et al. (2019) to suit the MT setting, with reference translations as the positive documents and the MT outputs as the negative documents. The results are shown in Table 10.

Table 10: Results of the re-trained coherence model.

| Training data | Test data | Accuracy |
|---|---|---|
| WMT11-15 | WMT17-18 | 77.35 |

**User study.** Figure 3 shows our user study interface. The participants are shown three candidate English translations of the same source text, and asked to rank the texts on how coherent and readable they are. To optimize annotation time, participants are only shown the first four sentences of the document; they annotate 100 such samples. We also include the reference as one of the candidates for control, and to confirm that we are justified in re-training the evaluation model to assign a higher score to the reference. Three participants took part in the study. Our control experiment results in an AC1 agreement of **0.84**.

The agreement between the human judgements and the rankings obtained by using the original coherence model trained on permuted WSJ articles (also news domain, like the WMT data), is 0.784. The fact that the original model performs no worse than 0.784 shows that there are definitely coherence issues in such (MT output vs reference) data that are being picked up.

The agreement between the human judgements and the retrained coherence evaluation model's rankings is **0.82**. Our re-trained model is therefore also learning useful task-specific features in addition

Table 11: Models **rank**ed according to their performance (best to worst) in coherence according our evaluation, with BLEU for comparison. Coherence scores given here are obtained by subtracting the score for the model translation from the score for the reference translation, and summing the score differences across the dataset. Hence, smaller model scores indicate better performance (closer to reference scores).

| De-En | | | | Ru-En | | | | Zh-En | | | |
|---|---|---|---|---|---|---|---|---|---|---|---|
| Rk | Model | BLEU | Score | Rk | Model | BLEU | Score | Rk | Model | BLEU | Score |
| 1 | CONCAT | 31.96 | 5038.057 | 1 | ANAPH | 27.66 | 7412.280 | 1 | ANAPH | 16.31 | 2599.465 |
| 2 | SAN | 29.32 | 5059.811 | 2 | CONCAT | 24.56 | 7641.254 | 2 | CONCAT | 17.17 | 2709.448 |
| 3 | S2S | 31.65 | 5120.633 | 3 | TGTCON | 26.06 | 7742.092 | 3 | S2S | 17.86 | 3545.707 |
| 4 | ANAPH | 29.94 | 5166.320 | 4 | SAN | 24.34 | 7875.095 | 4 | TGTCON | 15.76 | 4440.890 |
| 5 | TGTCON | 29.94 | 5475.636 | 5 | S2S | 23.88 | 8133.746 | 5 | SAN | 15.18 | 4538.013 |
| 6 | HAN | 29.69 | 5516.808 | 6 | HAN | 25.11 | 8719.370 | | | | |

to general coherence features. The high agreement validates our proposal to use the modified coherence model to evaluate the benchmarked MT systems.

**Results.** The total assigned scores (difference between reference and translation scores) obtained for each system after summing the over the samples in the respective testsets are given in Table 11. The models are ranked based on these scores from lowest score (best performing) to highest score (worst performing).

## A.3 LEXICAL CONSISTENCY

**Dataset extraction.** One possible issue with our method could be that reference translations may contain forced consistency, *i.e.,* human translators introduce consistency to make the text more readable, despite inconsistent word usage in the source. It may not be reasonable to expect consistency in a system translation if there is none in the source. To confirm, we conducted a manual analysis where we compared the lexical chains of nouns and adjectives in Russian source texts against the lexical chains in the English reference. We find that in a majority (77%) of the cases, the lexical chains in the source are reflected accurately in the reference, and there are relatively few cases where humans force consistency.

**Evaluation.** It is possible that the word used in the system translation is not the same as the word in the reference, but the MT output is still consistent (*e.g.,* a synonym used consistently). We tried to use alignments coupled with similarity obtained from ELMo (Peters et al., 2018) and BERT (Devlin et al., 2018) embeddings to evaluate such cases to avoid unfairly penalizing the system translations, but we found this to be noisy and unreliable. Thus, we check consistency against the reference; if at least one translation matches the reference but the translated word occurs fewer times than it does in the reference, the translation is considered inconsistent. For samples where there is no common translation between the system output and the reference, we cannot be sure if it is consistent or not, so we exclude it for calculating the primary ranking percentage. We therefore report both the percentage of consistent samples as a fraction of consistent + inconsistent samples, and percentage of consistent samples as a fraction of the full dataset for comparison purposes.

## A.4 DISCOURSE CONNECTIVES

Table 12: Results of the connective classification model.

| Training data | Test data | Precision | Recall | F1 score |
|---|---|---|---|---|
| PDTBv3 Sections 0-24 | PDTBv3 Sections 0-24 (avg CV) | 95.58 | 92.35 | 93.92 |

**Connective Classification model.** We build an explicit connective classifier to identify candidates that are acting in the capacity of a discourse connective. The model consists of an LSTM layer (Hochreiter & Schmidhuber, 1997) followed by a linear layer for binary classification, initialized by ELMo embeddings (Peters et al., 2018). We use annotated data from the Penn Discourse Treebank

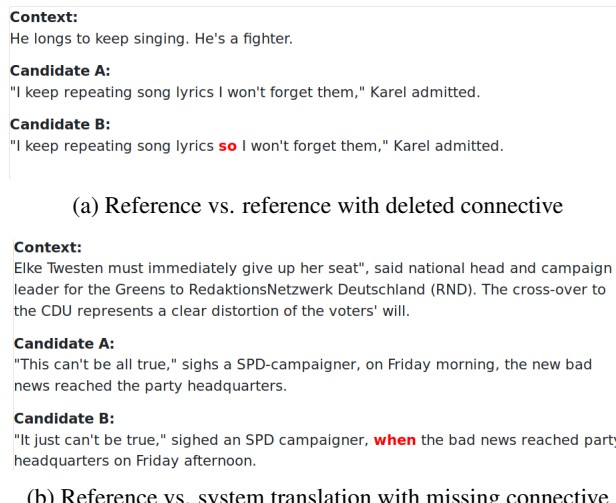

(a) Reference vs. reference with deleted connective

(b) Reference vs. system translation with missing connective

Figure 4: User study interfaces for discourse connectives.

(PDTBv3) (Prasad et al., 2018) and conduct cross-validation experiments across all 25 sections. The average Precision, Recall and F1 scores from the cross-validation experiments are reported in Table 12. Our classifier achieves an average cross-validation F1 of **93.92**, which shows that it generalizes very well. The high precision also provides certainty that the model is classifying discourse connectives reliably.

**User Study.**    To confirm that the presence of the connective conveys information and contributes to the readability and understanding of the text, we conducted two user studies. For the first study, as presented in Figure 4a, participants are shown two previous sentences from the reference for context, and asked to choose between two candidate options for the sentence that may follow. These options consist of the reference translation with the connective highlighted, and the same text with the connective deleted.

Participants are asked to choose the sentence which more accurately conveys the intended meaning. There were two participants who annotated 200 such samples. The reference with the connective was chosen over the version without the connective with an AC1 agreement of **0.98**. Table 13 shows the connective-wise breakdown.

In the second study, the participants were shown the reference along with the system translation that was missing the connective (Figure 4b). In this study, the setup has no artificially constructed data; the idea is to check if there is a possibility that the system translation is structured in such a way as to require no connective. However, the AC1 agreement for preferring the reference was **0.82** (2 annotators for 200 samples; different annotators from the first study) for this study as well, which is still quite high. Table 13 has the connective-wise breakdown; here we see that the results are slightly different for certain connectives, but overall the strong preference for the reference with the connective is retained. Our assumption that connectives must be translated is validated through both studies.

Note that participants may not prefer the version without the connective due to loss of grammaticality or loss of sense information. Although indistinguishable in this setting, we argue that since both affect translation quality, it is reasonable to expect a translation for the connectives.

Note that for both studies, participants were also given options to choose 'Neither' in case they didn't prefer either choice, or 'Invalid' in case there was an issue with the data itself (*e.g.,* transliteration issues, etc.); data that was marked as such was excluded from further consideration.

| Study 1: Reference vs. Connective Deleted Reference | | | | | Study 2: Reference vs. Missing Connective Translation | | | | |
|---|---|---|---|---|---|---|---|---|---|
| **Connective** | **AC1 Agr.** | **# Ref** | **# Noisy** | **# Tie** | **Connective** | **AC1 Agr.** | **# Ref** | **# Sys** | **# Tie** |
| and | 0.96 | 136 | 4 | 11 | and | 0.84 | 127 | 20 | 26 |
| also | 1.0 | 35 | 3 | 18 | also | 0.82 | 36 | 5 | 3 |
| when | 1.0 | 29 | 0 | 0 | when | 0.88 | 22 | 1 | 4 |
| after | 1.0 | 23 | 0 | 0 | after | 0.81 | 15 | 1 | 6 |
| by | 1.0 | 12 | 0 | 2 | by | 1.0 | 12 | 0 | 0 |
| or | 1.0 | 6 | 0 | 0 | or | -0.38 | 2 | 1 | 3 |
| as | 1.0 | 9 | 0 | 1 | as | 0.79 | 12 | 1 | 1 |
| while | 1.0 | 9 | 0 | 3 | while | 1.0 | 11 | 0 | 1 |
| so | 1.0 | 1 | 0 | 1 | so | 1.0 | 8 | 0 | 0 |
| because | 1.0 | 10 | 0 | 0 | because | 1.0 | 7 | 0 | 1 |
| then | 1.0 | 6 | 0 | 5 | then | 0.57 | 6 | 2 | 4 |
| with | 1.0 | 5 | 0 | 1 | with | 1.0 | 5 | 0 | 1 |
| if | 1.0 | 4 | 0 | 0 | if | 1.0 | 3 | 0 | 1 |
| thus | 1.0 | 2 | 0 | 0 | thus | 1.0 | 2 | 0 | 0 |
| indeed | 1.0 | 0 | 0 | 2 | indeed | 1.0 | 2 | 0 | 0 |
| still | 1.0 | 2 | 0 | 2 | still | 1.0 | 2 | 0 | 0 |
| without | 1.0 | 2 | 0 | 0 | without | 1.0 | 2 | 0 | 0 |
| unless | 1.0 | 2 | 0 | 0 | unless | 1.0 | 2 | 0 | 0 |
| until | 1.0 | 2 | 0 | 0 | until | 1.0 | 2 | 0 | 0 |
| therefore | -0.33 | 1 | 0 | 1 | therefore | 1.0 | 2 | 0 | 0 |
| subsequently | 0 | 0 | 0 | 2 | subsequently | 1.0 | 2 | 0 | 0 |
| ultimately | 0 | 0 | 0 | 2 | ultimately | 1.0 | 2 | 0 | 0 |
| before | 1.0 | 8 | 0 | 0 | before | -0.38 | 1 | 1 | 4 |
| previously | 0 | 0 | 0 | 2 | previously | 0 | 1 | 0 | 1 |
| once | 1.0 | 2 | 0 | 0 | once | 0 | 1 | 0 | 1 |
| however | 1.0 | 2 | 0 | 0 | however | 0 | 0 | 1 | 1 |
| in | 1.0 | 2 | 0 | 0 | in | - | - | - | - |

Table 13: Connective-wise results for the connective user studies. The table also shows the number of times the **Ref**erence / **Sys**tem translation was chosen (summed for both annotators). The **Tie** column shows the number of times the users showed no preference. Note that ties are not included in the agreement. Other samples not included were the ones marked as *invalid* by the annotators due to misalignment errors, severe grammatical issues, etc.

Table 14: Discourse phenomena: **Anaph**ora (restricted to anaphoric pronouns), **Lex**ical **Con**sistency, and Discourse **Conn**ectives in popular NMT datasets (for English). The column **ANY** shows the proportion of sentences which contain any of the listed phenomena.

| **Dataset** | **Anaph.** | **Lex. Con.** | **Conn.** | **ANY** |
|---|---|---|---|---|
| UN | 0% | 31% | 0% | 31% |
| Europarl | 17% | 24% | 12% | 49% |
| News Commentary | 5% | 18% | 18% | 37% |
| IWSLT | 11% | 19% | 32% | 42% |

## A.5 TGTCON MODEL ARCHITECTURE

Here we describe the model architecture for TGTCON. The decoder introduced in Vaswani et al. (2017) is used for encoding the target sentence and the encoder adopted from the original encoder of the Transformer architecture is used to encode context of the target side. Each part, target decoder and context encoder, is repeated 6 times ($N$=6). In the last layer, two multi-head attention operations are performed, followed by layer normalization (similar to Vaswani et al. (2017)). The first multi-head attention is the self-attention on target sentence, whereas the second multi-head attention is a cross attention between representation of target and target context. These two representations are fused by gated linear sum which decides the amount of information from each representation that is to be passed on. Figure 5 shows the model architecture in detail.

## A.6 MODEL TRAINING

**Training Data** It is essential to provide the models with training data that contains adequate amounts of discourse phenomena, if we expect them to learn such phenomena. To construct such datasets, we first manually investigated the standard WMT corpora consisting of UN (Ziemski et al., 2016), Europarl (Tiedemann, 2012) and News Commentary, as well as the standard IWSLT dataset (Cettolo et al., 2012). We analyzed 100 randomly selected pairs of consecutive English sentences from each dataset, where the first sentence was treated as the context. Table 14 shows the percentage of cases containing the respective discourse phenomena.

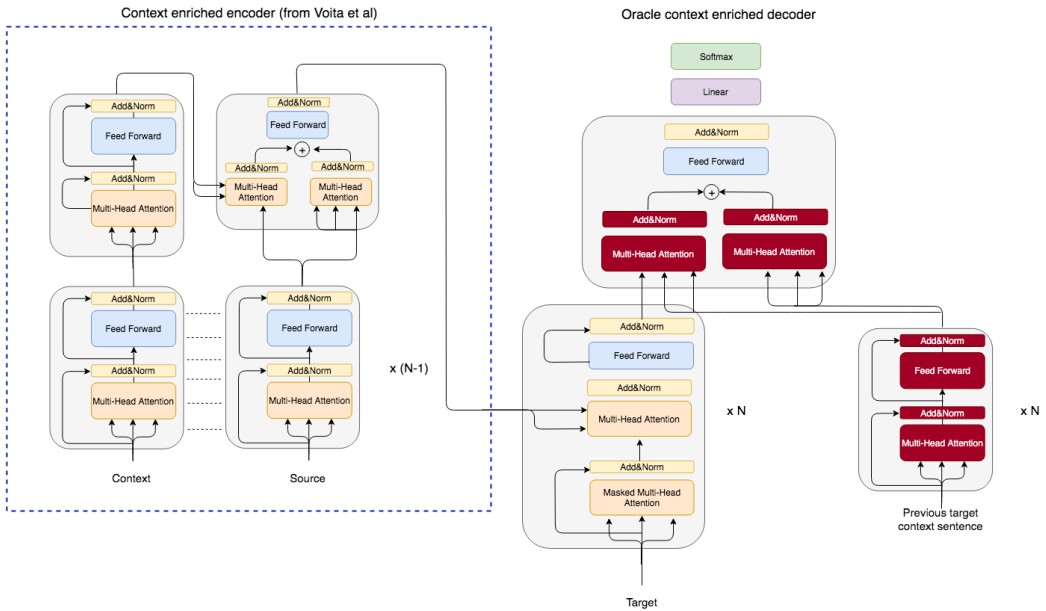

Figure 5: TGTCON model architecture.

In accordance with intuition, data sources based on narrative texts such as IWSLT exhibit increased amounts of discourse phenomena compared to strictly formal texts such as the UN corpus. On the other hand, the UN corpus consists of largely unrelated sentences, where only lexical consistency is well-represented due to the usage of very specific and strict naming of political concepts. We decided to exclude the UN corpus and combine the other datasets that have more discourse phenomena for De-En and Ru-En; for Zh-En, we keep UN and add WikiTitles to bolster the BLEU scores. Our training dataset is therefore a combination of Europarl, IWSLT and News Commentary datasets, plus UN and WikiTitles for Zh-En. The development set is a combination of WMT-2016 and older WMT data (excluding 2014). Note that our validation set does not have any data in common with the benchmark testsets. We test on WMT-2014 (De/Ru-En)/WMT-2017(Zh-En) data. We tokenize the data using Jieba for Zh and the Moses software[5] for the other languages, lowercase the text, and apply BPE encodings[6] from Sennrich et al. (2016). We learn the BPE encodings with the command `learn-joint-bpe-and-vocab -s 40000`.

**Training** For the context-aware models, we use the implementations from official author repositories. As the official code for ANAPH (Voita et al., 2018b) has not been released, we implement the model in the Fairseq framework (Ott et al., 2019).[7] For training the S2S and CONCAT models, we used the Transformer implementation from **fairseq**.We confirmed with the authors of HAN and SAN that our configurations were correct, and we took the best configuration directly from the ANAPH paper.

## A.7 MODEL PARAMETERS

Parameters used to train HAN are displayed in Table 15, and parameters for the S2S, CONCAT, ANAPH, and SAN models are displayed in Table 16.

## A.8 ERROR EXAMPLES

Examples for the different types of errors encountered across the tasks are given in Table 17.

---

[5]https://www.statmt.org/moses/
[6]https://github.com/rsennrich/subword-nmt/
[7]https://github.com/pytorch/fairseq

Table 15: Configuration parameters for training HAN model, taken from the authors' repository
`https://github.com/idiap/HAN_NMT/`

| Parameters | Values |
|---|---|
| **Step 1: sentence-level NMT** | |
| -encoder_type | transformer |
| -decoder_type | transformer |
| -enc_layers | 6 |
| -dec_layers | 6 |
| -label_smoothing | 0.1 |
| -rnn_size | 512 |
| -position_encoding | - |
| -dropout | 0.1 |
| -batch_size | 4096 |
| -start_decay_at | 20 |
| -epochs | 20 |
| -max_generator_batches | 16 |
| -batch_type | tokens |
| -normalization | tokens |
| -accum_count | 4 |
| -optim | adam |
| -adam_beta2 | 0.998 |
| -decay_method | noam |
| -warmup_steps | 8000 |
| -learning_rate | 2 |
| -max_grad_norm | 0 |
| -param_init | 0 |
| -param_init_glorot | - |
| -train_part sentences | - |
| **Step 2: HAN encoder** | |
| *others - see Step 1* | *others - see Step 1* |
| -batch_size | 1024 |
| -start_decay_at | 2 |
| -epochs | 10 |
| -max_generator_batches | 32 |
| -train_part | all |
| -context_type | HAN_enc |
| -context_size | 3 |
| **Step 3: HAN joint** | |
| *others - see Step 1* | *others - see Step 1* |
| -batch_size | 1024 |
| -start_decay_at | 2 |
| -epochs | 10 |
| -max_generator_batches | 32 |
| -train_part | all |
| -context_type | HAN_join |
| -context_size | 3 |
| -train_from | [HAN_enc_model] |

Table 16: Configuration parameters for training SAN, ANAPH, CONCAT, S2S models. Parameters of ANAPH are taken from the original paper Voita et al. (2018b) and parameters of SAN are taken from the authors' repository: `https://github.com/THUNLP-MT/Document-Transformer` and user manual for the THUMT library which provides the basic Transformer model: `https://github.com/THUNLP-MT/THUMT/blob/master/UserManual.pdf`. Parameters which are not listed were left as default. Note that the `max-update` for Zh-En was set to 400,000 due to the larger dataset size.

| Model | Parameters | Values |
|-------|-----------|--------|
| SAN | **Step1: sentence-level** | |
| | batch_size | 6250 |
| | update_cycle | 4 |
| | train_steps | 200000 |
| | **Step 2: context-aware Transformer** | |
| | num_context_layers | 1 |
| ANAPH | –optimizer | adam |
| | –adam-betas | '(0.9, 0.98)' |
| | –clip-norm | 0.0 |
| | –lr-scheduler | inverse_sqrt |
| | –warmup-init-lr | 1e-07 |
| | –warmup-updates | 4000 |
| | –lr | 0.0007 |
| | –min-lr | 1e-09 |
| | –criterion | label_smoothed_cross_entropy |
| | –label-smoothing | 0.1 |
| | –weight-decay | 0.0 |
| | –max-tokens | 1024 |
| | –update-freq | 32 |
| | –share-all-embeddings | - |
| | –max-update | 100000 |
| CONCAT | –optimizer | adam |
| | –adam-betas | '(0.9, 0.98)' |
| | –clip-norm | 0.0 |
| | –lr-scheduler | inverse_sqrt |
| | –warmup-init-lr | 1e-07 |
| | –warmup-updates | 4000 |
| | –lr | 0.0007 |
| | –min-lr | 1e-09 |
| | –criterion | label_smoothed_cross_entropy |
| | –label-smoothing | 0.1 |
| | –weight-decay | 0.0 |
| | –max-tokens | 4096 |
| | –update-freq | 8 |
| | –share-all-embeddings | - |
| | –max-update | 100000 |
| S2S | *as in* CONCAT | *as in* CONCAT |

Table 17: Examples for the types of errors found in the translations. **S:** denotes source, **T:** denotes model translations while **R:** denotes reference translations.

| Phenomenon | Example |
|---|---|
| **Anaphora** | |
| Gender Copy | S: *Mir wurde **diese Wohnung** in Earls Court gezeigt, und **sie** hatte ...* 
 T: I was shown this **apartment** in Earls Court , and **she** had .. |
| Named Entity | T: ... ***Lady Liberty** is stepping forward. **It** is meant to be carrying the torch of liberty* 
 R: **She** is meant to be carrying the torch of Liberty. |
| Language Specific | S: *Ihr Auftraggeber: Napoleon.*, the pronoun *ihr* refers to the noun *Karten* (English: *maps*). 
 The German pronoun *ihr* can mean *her*, *their*, or *your*. 
 T: *(..) detailed maps for towns and municipalities (...). **Your** contractor : Napoleon.* 
 R: (..) detailed maps for towns and municipalities (...). **Their** commissioner: Napoleon. |
| **Lexical Consistency** | |
| Synonym | T: *Watch the Tory party **conference**. The **convention** is supposed to be about foreign policy, (...).* 
 R: Under tight security - the Tory party **conference**. The party **conference** was to address foreign policy (...). |
| Related Word | T: *In the collision of the **car** with a taxi, a 27-year-old passer was fatally injured.* 
 R: A 27-year old passenger was fatally injured when the **ambulance** collided with a taxi. |
| Named Entity | T: *The **Feeding-Community** farmer , however , also had the ready-filled specialities.* 
 *The demand for the good "made in **Feed orf**" was correspondingly high.* 
 R: But the **Füchtorf** farmer also had bottled specialties with him. 
 There was a lot of demand for the good "made in **Füchtorf**" beverage. |
| Omission | T: *(...) during the single-family home attempt, it stayed by the royal highlands thanks to the burglar **alarm**.* 
 *They got off when the culprits turned hand on Friday just before 20 a.m.* 
 R: It is thanks to the **alarm** system that the attempt in the Königswieser Straße at the single family home (...). 
 On Friday just before 20.00 the **alarm** rang when the offenders took action. |
| **Coherence** | |
| Ungrammatical | T: *"They didn't play badly for long periods – like Stone Hages , like Hip Horst – Senser.* 
 *Only the initial phase, we've been totally wasted", **annoyed the ASV coach.*** 
 R: "Over long periods, they had - as in Steinhagen, as against Hüllhorst - not played badly. 
 We only overslept the initial phase", **said the ASV coach annoyed.** |
| Hallucination | T: *Before appointing Greece , Jeffrey Pyett was the US ambassador to Kiev.* 
 *When it came to the Maidan and the coup in 2014 , it was a newspaper.* 
 R: Before his appointment, Geoffrey Ross Pyatt was an ambassador in Kiyv. 
 During his mission, the Maydan events and state coup happened, reminds Gazeta.Ru |
| Inconsistency | T: *The one-in-house airline crashed on Sunday afternoon at a parking lot near **Essen-Mosquitos**.* 
 ***Essen Mill** is a small airport that's used a lot by airline pilots.* 
 R: On Sunday afternoon, the single-seated aircraft crashed (..) a parking lot near the airport **Essen-Mülheim** 
 **Essen-Mülheim** is a small airport, which is frequently used by pilots with light private planes. |
| **Discourse Connectives** | |
| Omission | T: *Two people died driving their car against a tree .* 
 R: Two people died **after** driving their car into a tree. |
| Synonym | T: *Naldo's "great sport" on the bank, **too**.* 
 R: *Naldo is **also** a great athlete on the bench* |
| Mistranslation | T: *Gfk's leadership departs from disappointing business figures* 
 R: GfK managing director steps down **after** disappointing figures |

