# OpenReview forum: "DiP Benchmark Tests: Evaluation Benchmarks for Discourse Phenomena in MT"
_ICLR.cc/2021/Conference — Reject_

### Official Review · AnonReviewer4 · 2020-10-26
**Valuable investigations into discourse phenomena but does not fit for ICLR**

**Rating:** 4
**Confidence:** 4

**Review:**

This paper presents four methods for creating benchmarking datasets, each focusing on a particular discourse phenomenon which is difficult/hard to solve with existing context-aware neural machine translation (NMT) models.  It also evaluates several existing NMT models using the created data and makes a remark about their current status, such as the superiority of one method in one task.  This work has the originality and I am happy to know that some researchers devote their efforts to address these issues.  Moreover, I am sure that the field of NMT definitely benefits from this work, once the created data are made publicly available.

However, I am very skeptical that the contribution of this paper fits for ICLR, since the methodology presented in this paper itself mainly consists of a pile of human efforts.  In other words, if I understand the contents properly, only the technical advancement presented in this paper is TgtCon, a variation of the anaphora-centric method proposed by Voita et al. (2018b).  However, it cannot be a substantial merit to the community, since it does not necessarily perform better than existing methods and the reason of its deterioration is not analyzed.

Considering the main focus of the paper, i.e., creating benchmarking datasets, the paper should rather fit for a journal article in the field of natural language processing, where the authors can give more details in creating the datasets, such as procedure, tool, and attributes of annotators, not in appendices, and more careful analyses of the results, including WHY each method does (not) perform well on a particular discourse phenomenon.

Indeed, when I was reading this paper, I suffered from the fact that the main part of the paper is not self-contained.  For instance, the proposed TgtCon is not explained in Section 2 but the readers are advised to see an appendix.  Other information that is indispensable in data creation but missing in this paper is the detail of human judgment, such as the proficiency of the evaluators, protocol, and judgment criteria.  Each section reports on agreement ratio, but not Cohen's kappa.  With these reasons, I am not completely convinced of the quality of the resulted datasets and the portability of the proposed methods.

Below shows some questions.

Q1. Is there a reason to choose these four particular discourse phenomena?  Are they exclusive to each other?

Q2. Test sets for coherence/readability and discourse connectives are significantly smaller than those for the other two phenomena.  I am not sure that a test set with 200 or less examples is enough.

Q3. In Section 3.2.1, the authors regard the inconsistency of named entities.  However, in many error classification schemes, terminology errors are, irrespective of incorrect translations and inconsistency, distinguished from coherence errors.  For instance, the Multidimensional Quality Metrics (*1) locate them in different branches.

(*1) http://www.qt21.eu/mqm-definition/

Q4. What does "Random translations" in Section 3.3.1 mean?

---

> ### Author Response · Authors · 2020-11-13
> **Thank you for your review! Addressing your concerns. (Part 1)**
>
> Thank you for recognizing the scalability and originality of our work. We address your concerns here.
>
> **Lack of technical advancement/Suitability of Venue:** We humbly disagree with your view! The main goal of the paper is to introduce extensive discourse based benchmarks for MT models and to motivate finer-grained evaluations of models across languages. To this end, we introduced several carefully constructed test sets and evaluation measures that were validated against human judgments.
>
> Although technical advancement is not the main contribution of this paper, it is the overarching objective of the paper. We aim to push for improvements in NMT models that go beyond merely optimizing word-overlap with reference texts. NMT models are evolving fast (and are commonly part of ICLR submissions) but there is a distinct lack of fine-grained testing. It is our contention that frameworks for evaluation of models must be introduced in the same venues as the models themselves, in order to ensure that better evaluation practices are adopted by the community that produces the models. For the same reason, papers on robustness of neural models have been well accepted by the ICLR community (e.g., [1] accepted as an oral). More closely related to ours, popular benchmarking frameworks such as GLUE [2] and benchmark for RL [3] were also introduced through ICLR. Similarly, the SuperGLUE [4] and XTREME [5] benchmarks were introduced in NeurIPS-2019 and ICML-2020, respectively. In this regard, please allow us to quote a common reviewing oversight referred from the ICLR website (https://iclr.cc/Conferences/2021/ReviewerGuide):
>
> *““This work does not fit topic X.” – When reviewers come across a paper that does not fit the typical paper mold for a particular venue, one common response is to dismiss it for this very reason. If you find yourselves writing this in your review, stop and think whether you are being needlessly conservative. Is it possible that this new approach will actually move the field forward?”*
>
> We sincerely hope that our reviewer reconsiders their decision and gives our paper a fair chance.
>
> **Agreements reported:** As mentioned in the Appendix (A1), agreements reported in the paper are the AC1 Gamma coefficient proposed by Gwet (2008). Due to the nature of the datasets used for the user studies (essentially erroneous translations vs reference), annotators are more likely to choose the reference as the better candidate. This yields a skewed distribution of the annotations; traditional correlation measures such as Cohen’s Kappa are not robust to this. Therefore we report the more appropriate Gwet’s AC1/Gamma coefficient. It is also the agreement reported by Jwalapuram et al. (2019).
>
> **Choice of discourse phenomena:** Yes, they are mostly exclusive to each other (although coherence has been used as a generic term to mean all the phenomena that contribute to document level understanding). In the literature, discourse phenomena have been broadly categorized into two: cohesion and coherence. Anaphora and lexical consistency fall under cohesion, while discourse relations (e.g., Contrast, Explanation) and connectives fall under coherence. These are the four main discourse phenomena; all other phenomena can be subsumed under one of these. Both Sennrich (2018) [6] and Hardmeier (2018) [7] list these four as the main discourse phenomena.
>
> --
>
> [1] Synthetic and Natural Noise Both Break Neural Machine Translation. Yonatan Belinkov, Yonatan Bisk. In ICLR 2018 (Oral presentation).
>
> [2] GLUE : A MultiTask Benchmark and Analysis Platform for Natural Language Understanding. Alex Wang, Amanpreet Singh, Julian Michael, Felix Hill, Omer Levy and Samuel R. Bowman. In ICLR 2019.
>
> [3] Behaviour Suite for Reinforcement Learning.  Ian Osband, Yotam Doron, Matteo Hessel, John Aslanides Eren Sezener, Andre Saraiva, Katrina McKinney, Tor Lattimore, Csaba Szepesvari Satinder Singh, Benjamin Van Roy, Richard Sutton, David Silver, Hado Van Hasselt. In ICLR 2020.
>
> [4] SuperGLUE: A Stickier Benchmark for General-Purpose Language Understanding Systems. Alex Wang, Yada Pruksachatkun, Nikita Nangia, Amanpreet Singh, Julian Michael, Felix Hill, Omer Levy and Samuel R. Bowman. In NeurIPS 2019.
>
> [5] XTREME: A Massively Multilingual Multi-task Benchmark for Evaluating Cross-lingual Generalization. J. Hu, Sebastian Ruder, Aditya Siddhant, Graham Neubig, Orhan Firat and M. Johnson. In ICML 2020.
>
> [6] http://homepages.inf.ed.ac.uk/rsennric/wnmt2018.pdf
>
> [7] https://ufal.mff.cuni.cz/mtm18/files/03-discourse-in-mt-christian-hardmeier.pdf

---

> > ### Author Response · Authors · 2020-11-13
> > **Thank you for your review! Addressing your concerns. (Part 2)**
> >
> > **Smaller sizes of some datasets:**
> > 1. The **coherence and readability** test sets are entire documents, so the actual size of the testsets has thousands of sentences - 272 documents (5,611 sentences) for De-En, 330 documents (4,427sentences) for Ru-En and 210 documents (3,050 sentences) for Zh-En. MT data with document-boundaries is also rare.
> > 2. The **discourse connective** test set is currently limited by the fact that they occur relatively less often, and that we do not have the means to accurately evaluate synonymous translations, limiting us to use errors caused by omissions to create the test sets. In future work, we hope to create a more robust evaluation measure for discourse connectives, which will help us expand the test set to include more errors.
> >
> > Note that we also restricted the WMT data used to 2016 and beyond in order to capture errors from predominantly neural models (previous submissions were mostly statistical and rule-based MT systems). As subsequent campaigns occur, our automatic test set generation methods make it easy to update our test sets to include more and more data, while also making them more challenging as systems improve. We hope our reviewer understands the challenges and appreciates our sincere efforts in this valuable research.
> >
> > **Terminology errors in coherence:** As described before, coherence is a broad notion that includes several concepts like grammaticality, cohesion, topic structures, inter-sentential relations and so on. Somasundaran et al (2014) use lexical consistency as a proxy to represent coherence and use it to measure coherence quality. Inconsistent translations of words break continuity and lead to problems in inferring sentence relations across texts, which is a coherence and readability issue. Our analysis in Section 3.2.1 meant to point this out as one of the sources of a lack of coherence in machine translation output.
> >
> > **Random Translations:** These are sentences which are so distorted that it is impossible to say what actually happened to the pronoun/connective/etc. in question. In rare cases, the quality of the translation is sometimes so poor that it can become hard to recognize what the elements in the translation refer to, and the output is essentially random with respect to the phenomena we are trying to analyse.

---

### Official Review · AnonReviewer3 · 2020-10-26
**commendable, large-scale test sets; are metrics targeted and trustworthy?**

**Rating:** 4
**Confidence:** 4

**Review:**

This paper presents a benchmark for discourse phenomena in machine translation. Its main novelty lies in the relatively large scale, spanning three translation directions, four discourse phenomena, and 150-5000 data points per language and phenomenon. A relatively large number of systems from previous work is benchmarked on each test set, and agreement with human judgments is measured.

positives:

- clearly, a lot of thought and effort has gone into the creation of the benchmarks, and this is the most diverse set of benchmarks for discourse phenomena in MT so far.

- extensive experiments with models from previous work, along with human analysis.

negatives:

- two of the four benchmarks, anaphora and coherence, are evaluated by neural models trained on WMT outputs, so the interpretation of scores is opaque, and their validity is unclear. Specifically, Jwalapuram et al. (2019) train a neural network to distinguish references from MT output based on the ELMo representations of pronouns, but in principle, this model can use signals other than the correctness of pronouns translation to make this distinction. Similarly, the model by Moon et al. (2019) was originally trained to distinguish real documents from randomly shuffled ones, and I can see how their complex neural network would then learn to rely on coherence features. However, this submission uses reference translations as positive, MT output as negative examples, so it again may learn to use features other than coherence for its decisions.

- also, I'm not fully convinced about the validity of the automatic discourse connective evaluation. According to the manual analysis, there is a large proportion of false negatives (synonymous translations flagged as errors), and rankings would change if synonyms were counted as correct. I was also not satisfied with the evidence that the omission of connectives is generally an error. The human study was a bit simplistic in that it just deleted connectives (although more changes might be needed to ensure grammaticality) or used noisy MT output rather than alternative human references. It also seems to have been monolingual. If the test set consists of examples with ambiguous or implicit discourse relations in the source, then it may actually be the right translation strategy to omit them. I'm worried that a benchmark that rewards explicitation and punishes leaving discourse relations implicit may set the wrong incentives.

- I was surprised by the low results (already in terms of BLEU) of some of the tested variants. Authors describe in great care their efforts to fairly reproduce previous work, including the use of original code and hyperparameters where possible, but I can't help but think that the models are suboptimally trained, and that statements about whether context-aware models consistently improve discourse phenomena are tainted by this.

recommendation:

I'm leaning negative on the current version of the paper and benchmark. I think the test sets have been carefully assembled, and along with the various types of models evaluated on them, this work has value. But before I'd recommend that the benchmarks actually be used in the field, authors would need to improve upon the evaluation scores used for anaphora, coherence, and discourse connectives and make sure they really are targeted towards the phenomena they claim to measure, and do not have large blind spots.

further questions and minor problems:

- did you perform early stopping? What was your stopping criterion?

- do you have an explanation why anaphora scores differ wildly between systems for ZH-EN (table 9)?

- the discourse connectives test sets are based on examples where the reference contains a connective, but MT output does not. What MT system was used? Do the respective source segments generally contain explicit discourse connectives, or are the respective discourse relations generally implicit in the source?

---

> ### Author Response · Authors · 2020-11-13
> **Thank you for your review! Addressing your concerns. (Part 1)**
>
> Thank you for noting the effort and value of our work. We address your concerns here:
>
> **Validity of Model Evaluation:**
> 1. Anaphora (Jwalapuram et al., 2019): Although the model is trained on reference vs. MT output, their paper shows results for testing on two settings:
>     * Reference vs. MT output, where the model is 90.69% accurate,
>     * Reference vs. noisy reference, where everything in the noisy version is identical except for the pronoun; here the accuracy is 89.11%.
> They also report high agreements (>=0.8) with human judgments on similar noisy data. This convinced us that the model focuses on pronouns during evaluation.
>
> 2. Coherence (Moon et al., 2019): Coherence is complex, and can include grammaticality, topic structures and inter-sentential relations, which the Moon et al. (2019) model claims to capture. We validated the model's ability to distinguish coherence (and readability) by conducting a user study, where we compared the model's ranking of texts against human rankings of texts and found strong agreements between the two (>0.8). Since coherence is a subjective concept, there is no other way to validate the effectiveness of the model other than by comparing to human judgments, which we therefore do.
>
> We hope the reviewer can understand the challenges and appreciate our sincere effort to address them, especially considering the fact that currently there is no such established benchmark to measure progress. As mentioned in our response to AnonReviewer1, there has been a recent trend towards model-based evaluation metrics like MoverScore (Zhao et al, EMNLP-2019), BERTScore (Zhang et al, ICLR-2020), BLEURT (Sellam et al, ACL-2020), etc.
>
> **Validity of Discourse Connectives test set:**
>
> **Evaluation:** Discourse connectives are complex, and can be quite hard to evaluate. It can be hard to perfectly define synonyms and distil them into a list, as they can convey different discourse relations depending on the context. In light of the fact that evaluating erroneous translations under such circumstances would be extremely difficult, we chose to instead concentrate on cases where MT systems omit the connective, since omissions are easier to detect and evaluate. We agree that synonyms should not be penalized; this is why we included the percentages for when ANY candidate connective was produced by the MT output in order to give them the benefit of the doubt.
>
> **User studies:** Our decision to conduct two user studies for the connective test set validation stemmed from our discussion with the creators of the PDTB corpus. Note that both the studies provide context information to the users.
> 1. The first study uses the reference with the connective deleted, compared to the reference with the connective. This study serves to prove that connectives serve a purpose and omitting them is problematic. The high agreement (>0.9) results of this study preferring the reference with the connective indicate that connectives are necessary for grammaticality or for discourse relations to hold.
> 2. The second study uses the MT output with the missing connective, compared to the reference with the connective. This study serves to prove that the MT outputs are indeed making omission errors. If the MT outputs with missing connectives were structured in such a way as to have implicit discourse relations, the agreements that favoured the references should be significantly lower; however we find strong (>0.8) agreements that favour the reference with the connective.
>
> In general, manual examination has shown that the source sentences do contain an explicit connective, and that reference translations very rarely introduce extra words that do not occur in the source (we also detail a similar examination conducted for lexical consistency in the appendix).
>
> In summary, as the first of its kind benchmark for discourse level phenomena in MT, we left no stone unturned to make the evaluation as trustworthy as possible.
>
> **MT systems used for testset generation:** As mentioned in Section 3, all testsets are based on multiple MT system outputs submitted to WMT, ensuring that there is no bias against particular types of models. For De-En, Ru-En and Zh-En, these consist of translation outputs from 68, 41 and 47 unique systems respectively.

---

> > ### Author Response · Authors · 2020-11-13
> > **Thank you for your review! Addressing your concerns. (Part 2)**
> >
> > **Low BLEU results/stopping criterion:**  As already noted, we took care to ensure that the parameters we used to train the MT models were correct; we confirmed with the authors of the models wherever possible that our configurations were right (in particular we did this for the HAN and SAN models). We picked the models with the best BLEU on the validation set as is standard practice.  One thing to note is that our training datasets are different from those used in the original model papers, so the BLEU scores can change. If authors feel their model received too low a score, they can submit their own results to the benchmark as it will be public and our BLEU results are not final in any way. We also note that authors use various configurations of BLEU; for example, in the SAN paper (Zhang et al., 2018) the Zh-En scores are over 40. However, in Xiong et al. (2020) [1], the Zh-En scores are on the scale of 20. When we compute BLEU-1 on our SAN translations, we also get ~40 as in the original SAN paper. For our final results we report BLEU-4.
> >
> > One of the main points we wanted to emphasize was the inability of BLEU scores to provide a complete understanding of model performance. Our evaluations also serve to point out the strengths in certain models that BLEU fails to capture. This is highlighted by the consistently strong performance of the ANAPH model in discourse phenomena for Zh-En despite not having a top BLEU score.
> >
> > **Difference in anaphora scores:** We did note the difference in the scores produced by the evaluation model between systems for Zh-En, particularly considering the BLEU scores do not vary significantly. We think this is because the systems do in fact vary widely in performance quality despite what the BLEU indicates. Note for example the exact same trend in the coherence scores (Table 11), which come from an independent model and dataset. Note also that our analysis shows a trend towards language specific errors in lower ranked models. Considering this, we hypothesized that the perceived translation quality is quite different from what the BLEU scores suggest. We also think this supports the validity of our evaluation models because they independently show a similar score trend (other similar trends in support include the consistently high performance of ANAPH for Ru-En; the model was originally optimized for En-Ru).
> >
> > We put in a lot of effort and took care to make our testsets and evaluation as complete and thorough as possible. We admit that they may not be absolutely perfect, but we believe that that is the inevitable nature of first steps in research. Flawed metrics such as BLEU and ROUGE continue to be widely used, often as the sole metric, to measure advances in models. It is our sincere endeavour to push research towards providing finer-grained results. Subsequent developments in better evaluation methods will lead to improvements in our framework as well.
> >
> >
> > [1] Xiong, Hao, Zhongjun He, Hua Wu, and Haifeng Wang. 2019. “Modeling Coherence for Discourse Neural Machine Translation”. Proceedings of the AAAI Conference on Artificial Intelligence 33 (01):7338-45.

---

> > ### Comment · AnonReviewer3 · 2020-11-21
> > **thanks for the response**
> >
> > Thanks for your thoughtful response. Some points of disagreement remain:
> >
> > - Jwalapuram et al. (2019) provide empirical support for their model's sensitivity (if there is a pronoun error, does the metric pick it up?). My concerns are about the model's specificity (if the metric ranks one output higher, can we be confident that this is because of a pronoun translation error?)
> >
> > - my point about Moon et al. (2019) is that their model captures coherence features not only (or even mainly) because of the neural architecture, but because of the training setup, where the model has to distinguish different permutations of a text. On the sentence level, these will be identical in terms of grammaticality, fluency, etc., so we have some guarantee that the model can only solve the training task based on coherence features. However, your negative examples are MT output, so there is no guarantee that the model actually learns to rely on coherence features. It might still be a well-performing metric similar to BERTScore, BLEURT etc., but I would like to see some support that it captures coherence more than other metrics. You performed a study with users, and report relatively high correlation; how is the correlation between the human rankings you obtained and other metrics?
> >
> > - you say that "[the human study on deleted discourse connectives] indicate[s] that connectives are necessary for grammaticality [...]". My point was that this may be true if you simply delete discourse connectives, but it would often be possible to create grammatical variants without connectives. For example:
> >
> > 1. "I keep repeating song lyrics **so** I won't forget them" (reference from figure 4a)
> > 2. "I keep repeating song lyrics I won't forget them" (deletion variant that you evaluated)
> > 3. "I keep repeating song lyrics; I won't forget them" (more plausible variant)
> >
> > I'm happy to agree that a lot of thought and effort went into this benchmark, and that a metric or benchmark does not need to be flawless to be useful. But this benchmark does not exist in a vacuum - other metrics for MT exist, including for discourse phenomena, and authors need to make the case that this benchmark is more meaningful for discourse phenomena than other metrics.

---

> > > ### Author Response · Authors · 2020-11-22
> > > **Thank you for your response. Addressing additional concerns**
> > >
> > > Thank you for going through our response and getting back to us. We address your additional concerns here:
> > >
> > > **Pronoun model specificity:** We have two reasons to be confident that the pronoun model’s rankings/scores result only from pronoun translation errors:
> > >
> > > 1. We conducted a user study for pronoun ranking explicitly to compare against model rankings (detailed in Appendix A1). We presented the users with the context and 3 candidate translations (one of which is the reference, as control), with the pronouns highlighted, and asked them to rank the texts in terms of their pronoun translations. We obtained an agreement of 0.91 between the human rankings from this study and the model rankings.
> > > 2. The original model architecture only uses pronoun representations in their final scoring layer. That is, the model first obtains contextual representations of the sentence and the pronouns in the sentence, then extracts *only* pronoun representations, which are then passed to the final layer that generates the scores. Jwalapuram et al (2019) also include attention heat maps in their paper that explicitly show that the scores are influenced only by the pronouns in the sentence, both for the reference vs. noisy and the reference vs. system translation cases.
> > >
> > >
> > > **Grammatical variants without connectives:**  We agree with this! This is where the second user study that uses the MT system outputs directly comes in; as described in our previous response (and also originally in Appendix A4):
> > >
> > > The second study uses MT outputs that are missing connective translations (that is, the data is not synthetically created by deleting a connective, but is originally missing in the system translation), compared to the references which include the connective translations. This study serves to prove that the MT systems are indeed making omission errors. If the MT outputs with missing connectives were structured in such a way as to have implicit discourse relations, the agreements that favoured the references should be significantly lower; however we find strong (>0.8) agreements that favour the reference with the connective.
> > >
> > > From Fig 4b,
> > >
> > > **Context:** “Elke Twesten must immediately give up her seat", said national head and campaign leader for the Greens to RedaktionsNetzwerk Deutschland (RND).
> > >
> > > **MT output:** “This can’t be all true”, sighs a SPD-campaigner, on Friday morning, the new bad news reached the party headquarters.
> > >
> > > **Reference:** “It just can’t be true”, sighed an SPD campaigner, **when** the bad news reached party headquarters on Friday afternoon.
> > >
> > > For your reference, we also present its source sentence with the connective:
> > >
> > > **Source:** "Das darf doch alles gar nicht wahr sein", seufzt ein SPD-Wahlkämpfer, **als** am Freitagvormittag die neue Hiobsbotschaft die Parteizentrale erreicht.
> > >
> > > **Coherence model reliance on coherence features:** One of the reasons why we chose to evaluate “Coherence *and* Readability” is that we wanted to encapsulate both text-level coherence and *other* issues in the translations (hence we name it coherence and readability), which is why we train on MT outputs vs. references.
> > >
> > > BERTScore and BLEURT are actually sentence-level metrics, so they cannot be used to rank at a document-level as we do with the coherence model. We did however check the agreements with the rankings obtained by using the original coherence model trained on permuted WSJ articles (also news domain, like the WMT data), which is 0.784. The fact that the original model performs no worse than 0.784 shows that there are definitely coherence issues in such (MT output vs reference) data that are being picked up. Our retrained model has an agreement of 0.828. This shows that our re-trained model is also learning useful task-specific features in addition to general coherence features, which forms the “readability” part of the evaluation.

---

### Official Review · AnonReviewer1 · 2020-10-29
**Progress towards better evaluation of context-aware NMT**

**Rating:** 7
**Confidence:** 3

**Review:**

In this paper, the authors propose specific test sets for document-level NMT. They target anaphora, coherence/readability, lexical consistency and discourse connectives, and are available for multiple language pairs. The first two challenge sets rely on model-based evaluation.

Strengths:

The test sets, which target various discourse phenomena, directly evaluate the output of the models (contrarily to some existing multiple-choice challenge sets).

The types of mistakes made by NMT models are manually examined.

The authors validate the quality of their metrics by comparing against human judgements (although the number of samples is arguably small).

Weaknesses:

The evaluated NMT models all date from 2018 or earlier.

The anaphora challenge sets are only a minor update over previous work.

All language pairs use English as the target language.

Other remarks and questions:

For the anaphora and coherence/readability test sets, future work may "cheat" by using the evaluation models as part of the NMT systems.

Is normalizing the scores actually useful? The reference scores should be the same across all systems, so it only shifts all results by a constant and doesn't affect relative performance.

Whats steps would be needed to construct similar test sets for En->X language pairs?

---

> ### Author Response · Authors · 2020-11-12
> **Thank you for your review! Addressing your concerns.**
>
>  Thank you for recognizing our contributions. We address some of the questions and concerns here:
>
> **All models from 2018 or earlier:** Context-aware models are not as common as sentence-level models although many MT researchers have advocated for it. We chose some well-established, popularly cited work since we thought they would be representative. We aimed at testing original author codes for maximum credibility, so we chose papers with implementations. We also picked some older models due to their properties (e.g. inclusion of target-side context).
>
> **Future work may “cheat”:** We agree that this is possible, but this would be true of any model-based evaluation. We cannot let this preclude us from using all model-based evaluations, since that would mean we would be limited to simple automatic metrics like BLEU that cannot produce finer-grained or nuanced evaluations (moreover, BLEU has also been optimized). Discourse phenomena, especially coherence, are complex concepts that cannot be captured by simple metrics, and require model-based evaluation. There has also been a recent trend towards model-based evaluation metrics like MoverScore (Zhao et al, EMNLP-2019), BERTScore (Zhang et al, ICLR-2020), BLEURT (Sellam et al, ACL-2020), etc. to name a few. We also think it would be non-trivial to use the evaluation models in NMT systems while balancing the translation objective. Moreover, for anaphora, the test set is independent of the evaluation model, so the error distributions are likely to be different.
>
> **Normalization:** The evaluation models are both pairwise ranking models, requiring reference and candidate translations as input together, and are trained to score the reference higher. Conceptually, it is not the absolute scores but the relative differences between pairs that are meaningful. Also, inputs with the same reference sometimes get slightly different scores due to some internal randomness stemming from ELMo embeddings. We added the normalization to account for these two issues; we will add this explanation to the paper.
>
> **Generalizability to other languages:** Our test sets can be generated in other languages (reproducing the answer above to AnonReviewer2 here for your convenience):
>
> 1. **Anaphora:** the pronouns need to be separate morphemes (and not attached to verbs etc.). If there are several equivalent pronoun translations, a list may be needed so they can be excluded from being considered translation errors. E.g. Miculicich Werlen and Popescu-Belis (2017) has such a list for French; a list can also be collected through user studies as in Jwalapuram et al (2019).
> 2. **Coherence:** The coherence model (Moon et al, 2019) used to find poorly translated texts was re-trained on reference vs. MT outputs, which is also certainly possible for other languages since WMT system outputs are available. The coherence model from Moon et al (2019) is an end-to-end neural model that does not rely on any language-specific features, and thus can be trained on any target language.  However, language-specific or multilingual coherence models could also be used since Moon et al (2019) primarily train and test their model on English (WSJ) data.
> 3. **Lexical Consistency**: A lemmatizer was used to reduce common suffixes for detecting lexical consistency (e.g. “box” and “boxes” should not be detected as inconsistent words), so a similar tool will be needed for any other target language. CLTK provides a lemmatizer for several languages.
> 4. **Discourse Connectives:** Discourse connectives also need to be separate morphemes. We trained a classifier trained on PDTB to identify connectives since they are ambiguous in English. Datasets analogous to PDTB in other languages e.g. PCC (German) and CDTB (Chinese) are available.
>
> We chose English as the target since we conducted extensive manual analysis and user studies, and it was the language that we had expertise available in.

---

> > ### Comment · AnonReviewer1 · 2020-11-23
> > **Thank you for your response**
> >
> > Thank you for your response.
> >
> > I still believe that additionally using more recent models would have made the paper more convincing. Nevertheless, your main contributions relate to evaluation, not model architectures, so this is not a critical issue for me. I would suggest nuancing your claim that "Surprisingly, we find that existing context-aware models do not improve discourse-related translations consistently across languages and phenomena."
> >
> > I agree that there is a place for model-based evaluation, which is why I didn't mention cheating as a weakness.
> >
> > I appreciate the details you provided on how your methods could be adapted to other target languages.

---

> > > ### Author Response · Authors · 2020-11-24
> > > **Paper updated based on suggestions**
> > >
> > > Thank you! We have now updated the abstract to say, *“Surprisingly, we find that the complex context-aware models that we test do not improve discourse-related translations consistently across languages and phenomena.”* We have also posted a comment that includes a summary of other changes made based on suggestions from the reviewers.

---

### Official Review · AnonReviewer2 · 2020-10-29
**Resources for discourse in translation**

**Rating:** 6
**Confidence:** 4

**Review:**

This paper presents a dataset, a trained evaluation metric and a leaderboard for evaluating discourse phenomena for machine translation. They test this on a range of discourse level translation models and develop metrics which evaluate the models according to their performance on four discourse phenomena: anaphora, lexical consistency, coherence and readability, and discourse connective translation.

Strengths:
This paper delivers multiple contributions which could have significant impact on the field of discourse level machine translation. They release data for three language pairs and using their method one could extend it relatively easily into others. I like the thoughtful way that the authors find examples of hard discourse phenomena and each phenomena requires distinct handling.

Weaknesses:
They rely on previous work to create the Anaphora test set and evaluation model (Jwalapuram et al. (2019)). They should have explained at a high level how the Jwalapuram evaluation model works and they should have given a general idea of the rules used to filter the anaphora test set: how many rules, an example, would this be possible to do for other languages or does it only work well for English?
I would have liked more discussion about the extensibility of their approach into languages other than English.
They should have used sacrebleu or at least specified if the BLEU scores were tokenised and true cased.

I think this paper should be accepted because of combined strength of the dataset/metric/leaderboard. I think fine grained evaluation of hard phenomenon is the way forward for improving already very good MT models.

---

> ### Author Response · Authors · 2020-11-12
> **Thank you for your review! Addressing your concerns.**
>
> Thank you for recognizing the strengths and impact of our contributions. We address some concerns here:
>
> **General rules used to generate the anaphora test set:**
> 1. We aligned the references with various MT system outputs from WMT & IWSLT campaigns.
> 2. Given a list of pronouns (a closed set), we find the cases where the pronoun translations in the references do not match the aligned, corresponding pronoun translations in the MT system outputs.
> 3. Jwalapuram et al (2019) provide a list of pronoun pairs that are not equivalent (non-interchangeable) translations of each other. We filter out the cases where the mismatched pronoun translations could be equivalent translations.
> 4. The corresponding source texts of the remaining wrongly translated cases are added to our test set.
>
> **Generalizability to other languages:** Our test sets can be generated in other languages, given certain simple conditions are met:
> 1. **Anaphora:** the pronouns need to be separate morphemes (and not attached to verbs etc.). If there are several equivalent pronoun translations, a list may be needed so they can be excluded from being considered translation errors. E.g. Miculicich Werlen and Popescu-Belis (2017) has such a list for French; a list can also be collected through user studies as in Jwalapuram et al (2019).
> 2. **Coherence:** The coherence model (Moon et al, 2019) used to find poorly translated texts was re-trained on reference vs. MT outputs, which is also certainly possible for other languages since WMT system outputs are available. The coherence model from Moon et al (2019) is an end-to-end neural model that does not rely on any language-specific features, and thus can be trained on any target language.  However, language-specific or multilingual coherence models could also be used since Moon et al (2019) primarily train and test their model on English (WSJ) data.
> 3. **Lexical Consistency:** A lemmatizer was used to reduce common suffixes for detecting lexical consistency (e.g. “box” and “boxes” should not be detected as inconsistent words), so a similar tool will be needed for any other target language. CLTK provides a lemmatizer for several languages.
> 4. **Discourse Connectives:** Discourse connectives also need to be separate morphemes. We trained a classifier trained on PDTB to identify connectives since they are ambiguous in English. Datasets analogous to PDTB in other languages e.g. PCC (German) and CDTB (Chinese) are available.
>
> **BLEU scores:** Our data was tokenized using standard Moses tokenization scripts (tokenizer.perl, normalize-punctuation.perl, etc.) for De/Ru/En and Jieba for Zh, with all text lowercased (De/En); the preprocessing steps are outlined in the Appendix (A6). The scores we reported were BLEU4, either computed through fairseq’s evaluation code or NLTK, which we verified against each other.

---

> > ### Comment · AnonReviewer2 · 2020-11-23
> > **changes in paper**
> >
> > Thank you for the detailed response. I recommend you add some of this information in the paper. Is this your plan?

---

> > > ### Author Response · Authors · 2020-11-24
> > > **Paper updated based on suggestions**
> > >
> > > Yes! We have now included the suggested details in the main paper and uploaded the updated version. A comment has been posted that summarizes the updates that we made based on other suggestions. Thank you!

---

### Author Response · Authors · 2020-11-24
**Updated paper: Additions to the main paper and Appendix based on Reviewer Suggestions**

Dear Reviewers and AC,

We have now uploaded a new version of the paper, in which the main paper and the Appendix have been updated to include some details and information that was suggested by the reviewers. Specifically, we include:

1. Clarification about the preprocessing and BLEU added to Section 2 (Datasets and Training) addressing AnonReviewer 2’s concerns.
2. Details about the user study conducted for the discourse connective testset in Section 3.4 (User Study) addressing AnonReviewer 3’s concerns.
3. A new section (Section 5) describing the generalizability to other languages based on our author response to AnonReviewers 1 and 2.
4. A detailed list of steps used to generate the anaphora test sets (Appendix A1) based on our response to AnonReviewer 2’s suggestions.
5. A brief description of the pronoun evaluation model and our arguments for the sensitivity and specificity of the model (Appendix A1) addressing concerns from AnonReviewers 2 and 3.
6. An explanation of the agreement statistics used (previously in a footnote), now added to the main text in Appendix A1 (User Study) to address AnonReviewer 4’s concerns.
7. The agreement for the original WSJ trained coherence model with human judgments, comparing the agreement with our retrained coherence model (A2). This is based on our author response to AnonReviewer 3.

We hope that our responses and the subsequent modifications to the paper address all of your concerns and that you now have a more positive outlook towards the paper.

---

### Decision · Program_Chairs · 2021-01-07
**Final Decision**

**Decision:**

Reject

**Comment:**

This paper introduces a dataset and a trained evaluation metric for evaluating discourse phenomena for MT. Several context-aware MT models are compared against a sentence level baseline. The paper develops metrics which evaluate the models according to their performance on four discourse phenomena: anaphora, lexical consistency, coherence and readability, and discourse connective translation. Data is released for three language pairs (all using English as the target language).

First, I’d like to point out that creating datasets and benchmarks for analyzing/evaluating discourse-level errors in machine translation is an extremely valuable contribution. This paper is addressing a very relevant problem and even though there is no new model/method/algorithm being proposed, this work *fits* this conference - it is my opinion that the community should welcome and value more than it currently does the efforts spent in creating high quality datasets that can help make progress in the field.

There was substantial discussion among reviewers about this paper.

The main weaknesses raised by the reviewers were:
- Limited information about the process to create the anaphora test, which was a contribution of prior work (Jwalapuram et al. (2019) - this was addressed in the updated version; but the anaphora challenge sets seem to be only a minor update over previous work.
- All language pairs use English as the target language, and it is not simple to extend this approach beyond English target languages.
- Lack of detail on how BLEU scores were computed (tokenised? true cased? My recommendation is to use sacrebleu) - this was clarified in the rebuttal.
- The evaluated NMT models all date from 2018 or earlier.
- Two of the 4 benchmarks (anaphora and coherence) are evaluated by neural models trained on WMT outputs, which makes the interpretation of scores is opaque, and their validity is unclear.

While the creation of a benchmark for discourse evaluation of MT is a laudable effort as mentioned above, it is my opinion that due to some of the weaknesses above the current version of this work is not yet ready for publication. However, I strongly encourage the authors to improve upon these points and resubmit their work to another venue. I list some suggestions below to improve this paper.

My biggest concern with the current version is the last weakness above. As pointed out by a reviewer, the framework of Jwalapuram et al. (2019) provides empirical support for the model's sensitivity (if there is a pronoun error, does the metric pick it up?). But they don’t necessarily capture model *specificity* (if the metric ranks one output higher, can we be confident that this is because of a pronoun translation error?). For the coherence metric, authors make an argument that their metric is sensitive to coherence issue, but concerns remain about whether it is sufficiently specific to these issues. In the rebuttal, authors argue that BLEURT is sentence-level, but they could easily aggregate sentence-level judgments and report correlation between BLEURT and human coherence  judgments to show that their metric correlates better with human coherence judgments than BLEURT or even just BLEU. Besides BLEURT, I would add there are other recently proposed metrics that may capture discourse phenomena (neural metrics trained against MQM annotations or sentence-level human assessments with document context) and should be compared against: check COMET [1] or PRISM [2] (the latter is sentence-based but could be adapted for paragraphs or documents).

There is also prior work comparing various context-aware machine translation approaches against a sentence-level baseline, some with negative findings [3,4,5]. I suggest the authors look at this related work in future iterations of their paper.

[1] https://arxiv.org/pdf/2009.09025.pdf

[2] https://arxiv.org/pdf/2004.14564.pdf

[3] https://www.aclweb.org/anthology/2020.eamt-1.24.pdf

[4] https://arxiv.org/pdf/1910.00294.pdf

[5] https://www.aclweb.org/anthology/2020.emnlp-main.81.pdf